# Seasonal mixed layer depth shapes phytoplankton physiology, viral production, and accumulation in the North Atlantic

Ben P. Diaz[1], Ben Knowles [1,11], Christopher T. Johns[1], Christien P. Laber[1,12], Karen Grace V. Bondoc [1], Liti Haramaty[1], Frank Natale[1], Elizabeth L. Harvey [2], Sasha J. Kramer [3,4], Luis M. Bolaños [5,13], Daniel P. Lowenstein [6], Helen F. Fredricks[6], Jason Graff [7], Toby K. Westberry [7], Kristina D. A. Mojica [7,14], Nils Haëntjens [8], Nicholas Baetge [9], Peter Gaube[10], Emmanuel Boss [8], Craig A. Carlson [9], Michael J. Behrenfeld[7], Benjamin A. S. Van Mooy [6] & Kay D. Bidle [1✉]

Seasonal shifts in phytoplankton accumulation and loss largely follow changes in mixed layer depth, but the impact of mixed layer depth on cell physiology remains unexplored. Here, we investigate the physiological state of phytoplankton populations associated with distinct bloom phases and mixing regimes in the North Atlantic. Stratification and deep mixing alter community physiology and viral production, effectively shaping accumulation rates. Communities in relatively deep, early-spring mixed layers are characterized by low levels of stress and high accumulation rates, while those in the recently shallowed mixed layers in late-spring have high levels of oxidative stress. Prolonged stratification into early autumn manifests in negative accumulation rates, along with pronounced signatures of compromised membranes, death-related protease activity, virus production, nutrient drawdown, and lipid markers indicative of nutrient stress. Positive accumulation renews during mixed layer deepening with transition into winter, concomitant with enhanced nutrient supply and lessened viral pressure.

[1] Department of Marine and Coastal Sciences, Rutgers University, New Brunswick, NJ 08901, USA. [2] Department of Biological Sciences, University of New Hampshire, Durham, NH 03824, USA. [3] Interdepartmental Graduate Program in Marine Science, University of California, Santa Barbara, Santa Barbara, CA, USA. [4] Earth Research Institute, University of California, Santa Barbara, Santa Barbara, CA, USA. [5] Department of Microbiology, Oregon State University, Corvallis, OR, USA. [6] Department of Marine Chemistry and Geochemistry, Woods Hole Oceanographic Institution, Woods Hole, MA 02543, USA. [7] Department of Botany and Plant Pathology, Oregon State University, Corvallis, OR 97331, USA. [8] School of Marine Sciences, University of Maine, Orono, ME 04469, USA. [9] Department of Ecology, Evolution, and Marine Biology, Marine Science Institute, University of California Santa Barbara, Santa Barbara, CA 93106, USA. [10] Applied Physics Laboratory, University of Washington, Air—Sea Interaction and Remote Sensing Department, Seattle, WA, USA. [11]Present address: Department of Ecology and Evolutionary Biology, University of California, Los Angeles, Los Angeles, CA, USA. [12]Present address: Centre for Ecology and Evolution in Microbial Model Systems, Linnaeus University, SE-39 182 Kalmar, Sweden. [13]Present address: School of Biosciences, University of Exeter, Exeter, UK. [14]Present address: Division of Marine Science, School of Ocean Science and Engineering, The University of Southern Mississippi, Stennis Space Center, Kiln, MS, USA. ✉email: bidle@marine.rutgers.edu

Phytoplankton are dynamic microbial primary producers that inhabit the sunlit regions of the ocean. Comprising less than 1% of total global biomass[1] and accounting for ~50% of global primary productivity[2], these unicellular photo-autotrophs are estimated to accumulate and turnover weekly[3]. Given the integral role of phytoplankton in marine food webs and global biogeochemistry, the physical and ecological drivers of bloom development and decline have been studied and char-acterized in several key models[4]. Central to many of these models is the relationship of phytoplankton cell physiology to the depth of the mixed layer—the uppermost region in the water column which is homogenized by convective and turbulent mixing.

Given constant sunlight, the mixed layer depth (MLD) largely dictates the daily availability of light to phytoplankton. The cri-tical depth hypothesis posits that, as the MLD shallows in the spring, phytoplankton photosynthetic rates increase in response to the increasing availability of daily irradiance. A critical MLD exists where photosynthetic rates can overcome respiratory losses, allowing for increased division rates and biomass accumulation[5]. Following this view, bloom initiation requires that photosynthetic rates exceed respiration, and that division rates exceed destructive loss rates, which are treated as constant throughout the year in the critical depth hypothesis formulations[5].

A different picture emerges from analysis of a 9-year satellite record of phytoplankton biomass in the subarctic Atlantic, which showed that depth-integrated phytoplankton biomass accumula-tion begins in late winter as MLDs deepened and division rates decreased. Positive accumulation rates continued throughout spring independent of division rates[6]. These findings led to the disturbance-recovery hypothesis, where biomass accumulation is initiated by deep mixing, which dilutes and decouples phyto-plankton growth from predatory losses. Continued accumulation in the spring is thought to be sustained by accelerations in division rate[4]. Thus, the disturbance-recovery hypothesis highlights the importance of loss processes in bloom dynamics and emphasizes the need to better characterize these processes in relation to MLD.

Intracellular response pathways within phytoplankton under-pin growth and death and may drive seasonal accumulation rates and bloom progression. Consequently, a suite of biomarkers has been developed over the past two decades to characterize and diagnose stress responses in phytoplankton communities. Phytoplankton cells are known to activate intracellular stress and autocatalytic programmed cell death (PCD) pathways, which include specific protease proteins such as caspases and metacas-pases. These proteins are activated in response to both abiotic stress and viral infection[7]. Macronutrient-limitation[8,9] and virus infection[10,11] can elevate specific lipid abundances within phy-toplankton cells including the neutral storage lipid triacylglycerol (TAG). Nutrient stress[12], viral infection[13], or light stress[14] can also increase intracellular reactive oxygen species (ROS) and reactive nitrogen species[13]. Elevated intracellular ROS can oxidize various cellular components, including a common membrane lipid, phosphatidylcholine (PC), leading to the formation of oxidized phosphatidylcholine[15] (OxPC). In turn, OxPC can destabilize membranes and lead to protein misfolding or cell death[16]. Other biomarkers are diagnostic of compromised cell membrane integrity (dead cells)[13,17,18], viral production, seasonal accumulation of dissolved organic carbon ($DOC_{SA}$)[19], and par-ticle aggregation/sinking/ascending dynamics (transparent exo-polymers, TEP)[20,21] (Supplementary Table 1). Each of these biomarkers represent cellular pathways or extracellular signatures associated with reduced net growth and biomass. To date, these biomarkers have not been examined in the context of changing MLDs and the disturbance-recovery hypothesis.

Here, we characterize the in situ physiological state of com-munities associated with four distinct phases of the phytoplankton annual bloom cycle, as part of the North Atlantic Aerosol and Marine Ecosystems Study (NAAMES)[22]. We show that each bloom phase had discernable signatures of ROS, OxPC, cell membrane integrity, PCD protease activity, and TAGs. These intracellular stress and death signatures were most distinctly pronounced in the Climax phase, when mixed layers recently shallowed, and in the Decline phase, with stable and shallow mixed layers and pronounced stratification. These two phases were also associated with elevated levels of extracellular bio-markers of stress and death, including TEP, viruses, caspase activity, and $DOC_{SA}$. Physiological trends were generally con-served among phytoplankton communities across a wide range of latitudes and subregions within a season. Shifts in these physio-logical biomarkers were observed within the same water column in response to mixing and prolonged stratification on the time-scale of 7 days. Lastly, physiological differences were discernable between sampling locations with distinct mixed layer depth his-tories within a bloom phase. Our findings provide population-level physiological context for the relationship between cellular loss and stratification in the North Atlantic, which support the disturbance-recovery hypothesis[23].

## Results and discussion
**Mixed layer depth and phytoplankton accumulation dynamics in the North Atlantic.** The NAAMES expeditions intensively measured biological, chemical, and physical properties from 4 to 7 locations, or stations, in each bloom phase during November (Winter Transition), March−April (Accumulation), May (Climax; same as Climax Transition[22]), and September (Decline)[22]. Stations spanned a broad range in latitude (~37 °N to ~55 °N, Fig. 1a), sub-regional classifications (Gulf Stream and Sargasso Sea, Subtropical, Temperate and Subpolar)[24], and MLDs (tens to hundreds of meters) (Fig. 1b and Supplementary Fig. 1). MLDs were calculated using a density difference threshold of 0.03 kg m$^{-3}$ from the top 10 m[25]. Field data and associated analyses are derived from phy-toplankton 1–20 μm in diameter and their associated communities sampled within the photic zone (40, 20, 1% surface irradiance) and within the mixed layer, unless otherwise noted.

Predominantly positive rates of phytoplankton accumulation were observed during three of the four phases of the annual bloom cycle (Fig. 1c, d and Supplementary Fig. 2b), including Winter Transition, consistent with prior observations[6,26]. The Decline phase had the lowest accumulation rates and the highest proportion of net phytoplankton loss rates (Fig. 1c, d). Accumulation rates were statistically indistinguishable ($p > 0.05$, Kruskal−Wallis) between populations collected from 5 m in-line sampling throughout the day (in situ) and contemporaneous incubations of the same phytoplankton populations under simulated in situ irradiance and temperature (incubations; see 'Methods') (Fig. 1c, d). Accumulation rates using incubations calculated via cell concentration or via biovolume were not statistically different (Supplementary Fig. 2b).

Phytoplankton cell concentration and biovolume generally increased with water column stability (stratification), during the Winter Transition, Accumulation, and Climax phases (Fig. 1e and Supplementary Fig. 2c). Stratification was quantified by the buoyancy frequency averaged over the upper 300 m of the water column (see 'Methods'). Higher values of buoyancy frequency indicate a more stratified water column where exchange with nutrient-rich water below the surface is reduced. Strongly stratified water columns (buoyancy frequencies above $2 \times 10^{-5}$ s$^{-1}$) during the Decline phase were associated with lower cell concentrations (Fig. 1e), consistent with enhanced phytoplankton loss or reduced accumulation. Phytoplankton biovolume and cell size distribution within 1–20 μm-sized phytoplankton cells increased during the

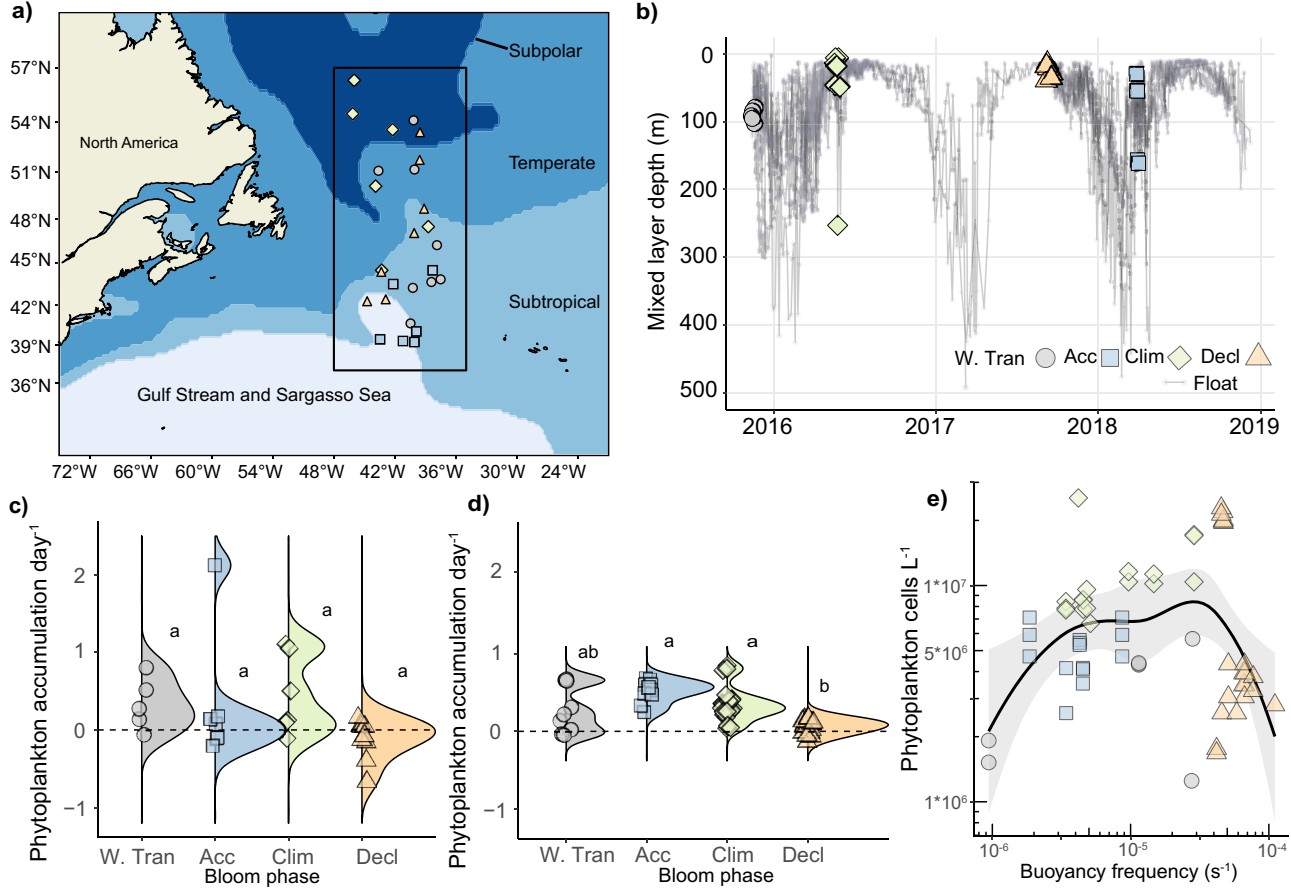

**Fig. 1 Mixed layer depth and phytoplankton accumulation dynamics. a** Locations of sampled stations within subregions of the Northwest Atlantic during the NAAMES expeditions (color coded and shaped by the bloom phase; W. Tran = Winter Transition; Acc = Accumulation; Clim = Climax; Decl = Decline; See key in Panel B). Black rectangle represents the study area of NAAMES and this research. **b** Mixed layer depths within the NAAMES campaigns (black box in Fig. 1a), calculated from CTD casts at each of the station locations (colored symbols) and Bio-ARGO profiling floats that were deployed at stations and sampled continuously (small circles with separate grey lines for each float). The latter provided a history of mixed layer depths before, during, and after occupation. **c** Bloom phase distribution of accumulation rates for in situ phytoplankton populations sampled several times per day at 5 m. Each point represents the median accumulation rate of each station. **d** Bloom phase distribution of phytoplankton cell accumulation rates derived from on-deck incubations of phytoplankton populations at simulated in situ light and temperature conditions (see 'Methods'). Each point represents a biological replicate. Data in panels (**c**) and (**d**) are based on cell concentrations and contoured with ridgeline smoothing to represent the distribution of accumulation rates across stations within a given bloom phase. The size of contour peaks is driven by frequency of observations. **e** Phytoplankton concentration (taken from 5 m) as a function of water column stratification (expressed as buoyancy frequency; $s^{-1}$). Higher buoyancy frequencies to the right of the plot represent more stratification. A LOESS line of best fit (shaded area = 95% confidence interval) for data shows the general trend of phytoplankton concentration across all seasonal phases. Different letters denote statistically significant groups ($p < 0.05$, Kruskal–Wallis test with Dunn corrections for multiple comparisons). Intergroup comparisons with more than one letter denote no significant difference between the two groups. Similar analyses as those presented in panels (**c**)–(**e**) are presented for phytoplankton biovolume in Supplementary Fig. 1. Exact $p$ values and number of biological replicates can be found in Source Data file.

Decline phase (Supplementary Fig. 2c–e). These higher biovolumes could have been a result of changes in community composition. They could have also been attributed to aggregation caused by virus infection[20,21,28], as virus concentrations were highest during this season (discussed below), or by light stress[27], as mixed layer populations were more consistently exposed to daily higher irradiance levels characteristic of shallow mixed layers (Fig. 1e).

In situ phytoplankton cell concentrations increased from Winter Transition until the Climax phase, from ~$1 \times 10^6$ to $2.5 \times 10^7$ cells $L^{-1}$ (Fig. 2a, c, gray boxes). On-deck incubations showed similar trends but had higher overall cell concentrations (Fig. 2a, c, white boxes). The Decline phase was characterized by a 4-fold reduction in median phytoplankton cell concentrations

from the peak abundances observed during Climax phase (Fig. 2a, c). The stress markers utilized in this study provided a unique view into the physiological status of communities across these annual bloom phases (Supplementary Table 1). Our ROS and compromised cell membranes biomarkers specifically targeted eukaryotic phytoplankton, given the conditions used for flow cytometry analysis (see 'Methods'). PCD-related proteases and lipids were extracted from biomass collected onto 1.2 and 0.2 μm diameter membrane filters, respectively. Consequently, these biomarkers could also include eukaryotic heterotrophs and bacteria in the system. Induction of caspase and metacaspase activities have been found in diverse phytoplankton, such as coccolithophores, diatoms, chlorophytes, nitrogen-fixing cyanobacteria, and dinoflagellates cells undergoing stress, senescence,

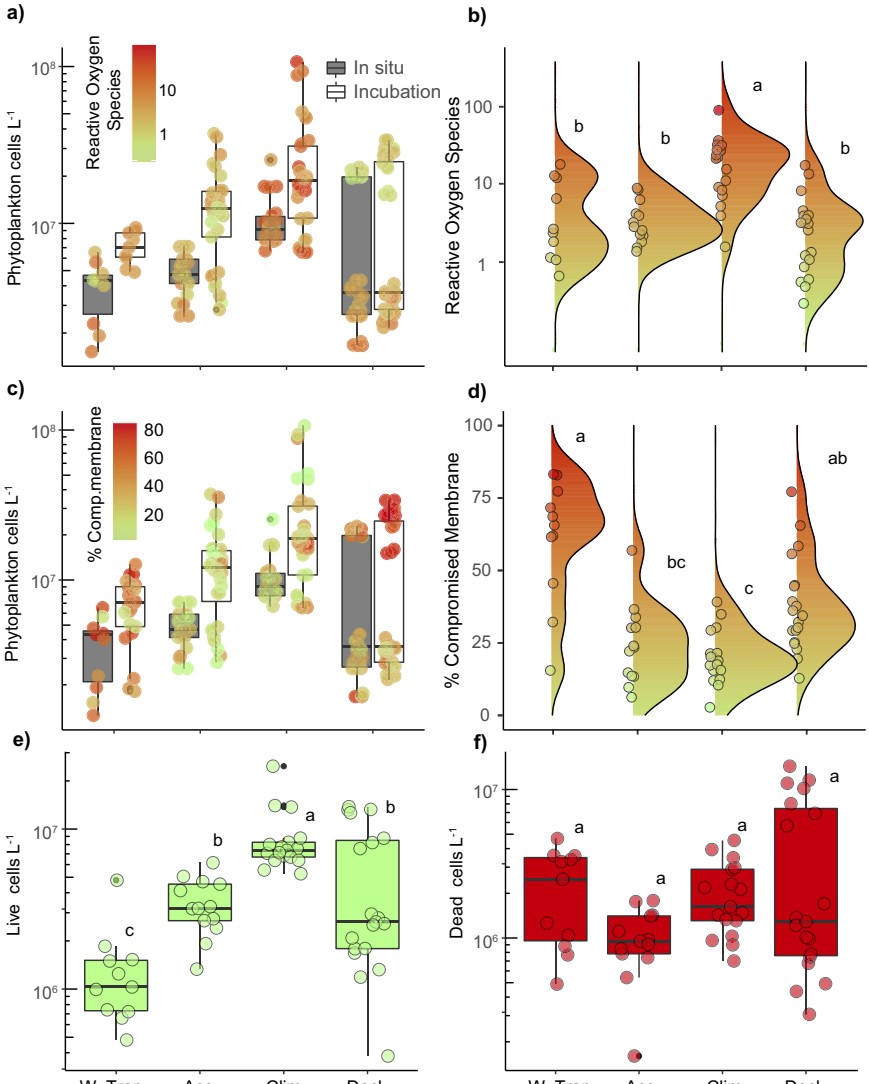

**Fig. 2 Seasonal phases have distinct physiological state signatures. a, c** Concentration of phytoplankton cells sampled within the mixed layer at depths associated with 40, 20, or 1% surface irradiance during different seasonal phases (W.Tran = Winter Transition; Acc = Accumulation; Clim = Climax; Decl = Decline). Data are shown for in situ water (grey bars) and on-deck incubations (open bars). Population-wide levels of **a, b** cellular reactive oxygen species (colored by fluorescence fold change from unstained; median per population) and **c, d** cell death (colored by % compromised membrane). Plots (**b**) and (**d**) are contoured with ridgeline smoothing to represent the relative in situ distribution of biomarker levels within each phase. The size of contour peaks is driven by frequency of observations. **e, f** In situ inventories of live (**e**; green) and dead (**f**; red) cells within the mixed layer through the different phases. Individual circles denote biological replicates. Box plots in (**a**), (**c**), (**e**) and (**f**) represent the median value bounded by the upper and lower quartiles with whiskers representing median + quartile × 1.5. Different letters denote statistically significant groups ($p < 0.05$, Kruskal−Wallis test with Dunn corrections for multiple comparisons). Intergroup comparisons with more than one letter denote no significant difference between the two groups. Number of biological replicates, by bloom phase/cast type (from left to right) = 10/9, 13/33, 17/36, 19/36 (**a**), 11/25, 13/36, 17/36, 19/36 (**c**), 11, 13, 17, 19 (**e, f**). Exact *p* values can be found in Source Data file.

and death[29]. They have also been reported in stressed or dying grazers[30], although no marine species has been explicitly studied. TAGs are found mainly in marine eukaryotic phytoplankton[31–33] and grazers[34]. The highly unsaturated fatty acids in the PC and OxPCs detected in our measurements are also indicative of eukaryotic organisms, and not marine cyanobacteria[32] or heterotrophic bacteria[35].

Collectively, these markers provide a broad interpretive context of the physiological state of the microbial system in response to changes in mixed layer depths throughout the North Atlantic bloom. We point out that some of our metrics (e.g., ROS, lipid biomarkers, PCD proteolytic enzymes) are not individually diagnostic of a particular stress or death process, such as virus

infection, nutrient limitation, or photo-damage. Instead, they represent core cell stress and death processes that are shared among biotic- and abiotic-stressors and are explicitly tied to cell fate, making them useful diagnostic indicators of physiological changes associated with bloom communities in relation to MLD and stratification.

Populations in the Accumulation phase had low ROS and few compromised cell membranes, indicative of healthy cells (Fig. 2b, d). Conversely, populations in the Climax phase were characterized by much higher ROS levels, diagnostic of oxidative stress (Fig. 2b). ROS production in phytoplankton has been linked to light stress[36,37], nutrient limitation[38,39], and virus infection[13]. The elevated ROS signatures during Climax phase were likely not

attributable to macronutrient limitation (Supplementary Fig. 3). Nutrient levels throughout NAAMES differed based on sub-regional water type, with Subpolar stations having the highest concentrations in the Climax phase ($NO_3 + NO_2 > 5\,\mu M$; $PO_4 > 0.4\,\mu M$). Notably, nutrient concentrations during the Climax phase were similar or higher than those observed for Accumulation phase samples, which had lower ROS signatures (Fig. 2b).

Phytoplankton cells in the Decline and Winter Transition phases had a higher percentage of compromised cell membranes, reaching levels as high as 80% (Fig. 2c, d). Both late stage viral infection and PCD have been linked to high levels of compromised membranes[13,29]. The percentage of phytoplankton cells with compromised membranes was used to calculate concentrations of live and dead cells within the mixed layer across the bloom phases. Living phytoplankton cell concentrations generally increased from the Winter Transition through the Climax phase (Fig. 2e). The variability of dead cells was highest in the Decline phase, which also had the largest variation in total, living, and dead cell concentrations (Fig. 2c, e, f).

Targeted analysis of OxPC, and TAGs in resident phytoplankton communities provided further context of changes in physiological states due to their relevance in cellular stress and loss processes. The seasonal bloom phases were characterized by distinct levels of these lipids (Fig. 3 and Supplementary Fig. 4). OxPC levels were highest in the Climax phase (Fig. 3a), where mixed layers had recently shallowed (Fig. 1b) and were concomitant with high intracellular ROS levels (Fig. 2b). Sub-cellular environments lacking in adequate antioxidant capacity are expected to accumulate OxPC[40] particularly when a shallow mixed-layer enhances UV exposure[15]. Chlorophyll-normalized TAG was highest in the Decline phase (Fig. 3b), which also had the lowest accumulation rates (Fig. 1c, d). High cellular TAG levels have been observed in senescent[41,42] or nutrient limited[9] diatoms, and virus infected haptophytes[43].

Caspase and metacaspase catalytic activity in community protein extracts provided additional subcellular diagnostics of stress and PCD activation[29]. Unlike metacaspase activity, which was detected at all depths and stations sampled, the incidence of caspase activity was lowest in the Accumulation phase (20%), increased in prevalence through the Climax (58%) and Decline (86%) phases, and then remained relatively high into the Winter Transition (59%) (Fig. 3c, d; pie graphs). Caspase-specific activities showed a similar pattern, being lowest in the Accumulation phase and highest in the Decline and Winter Transition phases (Fig. 3c). Caspase activity has been extensively linked to viral infection and autocatalytic PCD pathways in diverse eukaryotic phytoplankton[29,44,45]. Metacaspase-specific activities were highest in the Accumulation and Decline phases (Fig. 3d), which supports reported associations with normal cell function[46], nutrient stress, and viral infection[46,47] in unicellular eukaryotic phytoplankton.

**Stress, death, and extracellular signatures of loss and removal.** Given that TEP accumulation has an established mechanistic link with ROS and PCD activation in diverse marine phytoplankton[29], we analyzed concentrations of TEP in each phase of the annual biomass cycle. Both TEP $L^{-1}$ and TEP $cell^{-1}$ were highest during the Climax phase (Fig. 4a and Supplementary Fig. 5a), consistent with elevated cellular ROS levels (Fig. 2a, b) and a weakly stratified water column (Fig. 1e). TEP production was also highest in the Climax phase (Supplementary Fig. 5b), indicative of nutrient stressed or viral infected diatoms[48,49], chlorophytes[50], and haptophytes[21,28,49]. As cells[51] and viruses[52] in high TEP environments have altered aggregation, sinking, and ascending[53] potentials, their physical removal may be altered in this phase.

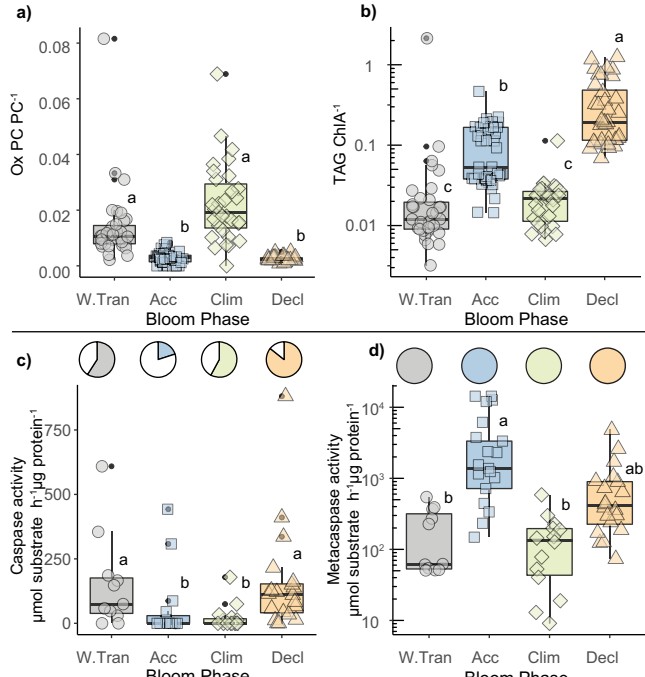

**Fig. 3 Seasonal phases are characterized by distinct lipid profiles and cell death-associated proteolytic activity. a** Oxidized phosphatidylcholine (OxPC40:10, OxPC42:11, OxPC44:12) normalized to total phosphatidylcholine (PC40:10, PC42:11, PC44:12). **b** Triacylglycerol (TAG; $pmol\ L^{-1}$), normalized to ChlA (peak area/L). **c** (top) The proportion of in situ samples with positive caspase activity (cleavage of IETD-AFC; color shading). (bottom) Caspase-specific activity rates ($\mu mol$ substrate hydrolyzed $h^{-1}\ \mu g\ protein^{-1}$) for in situ populations. **d** (top) The proportion of in situ samples with positive metacaspase activity (cleavage of VRPR-AMC; color shading). (bottom) Metacaspase-specific activity rates ($\mu mol$ substrate hydrolyzed $h^{-1}\ \mu g\ protein^{-1}$) for in situ populations. All box plots represent the median value bounded by the upper and lower quartiles, with whiskers representing median + quartile × 1.5. Different letters denote statistically significant groups ($p < 0.05$, Kruskal−Wallis test with Dunn corrections for multiple comparisons). Intergroup comparisons with more than one letter denote no significant difference between the two groups. Lipids and enzyme activities derived from biomass collected within the mixed layer at depths associated with 40, 20, or 1% surface irradiance (see 'Methods'). Individual symbols in all panels represent biological replicates and are colored and shaped by bloom phase. Seasonal phases are indicated in each panel (W.Tran = Winter Transition; Acc = Accumulation; Clim = Climax; Decl = Decline). Number of biological replicates, by bloom phase (from left to right) = 35, 37, 31, 36 (**a**), 35, 41, 36, 35 (**b**), 11, 15, 15, 20 (**c**), 11, 17, 14, 20 (**d**). Exact $p$ values can be found in Source Data file.

Coincident with elevated virus concentrations was an increase in seasonally accumulated dissolved organic carbon ($DOC_{SA}$). Viral lysis is an ecosystem process that leads to DOC accumulation[50,54,55] and can stimulate bacterial accumulation[56]. Virus and $DOC_{SA}$ concentrations were greatest during the Decline phase and varied by ~1000% and ~100%, respectively, across the annual cycle (Fig. 4c, e). We note that viruses and bacteria are operationally included in DOC measurements (see 'Methods'). However, calculations using conversions for average carbon quotas for virus particles (0.2 fg[57]) and bacterial cells (12.4 fg[58]) and their respective concentrations confirmed negligible contributions to total DOC and $DOC_{SA}$ concentrations (Supplementary Fig. 6; see 'Methods'). Virus and $DOC_{SA}$ concentrations were both positively correlated with water column

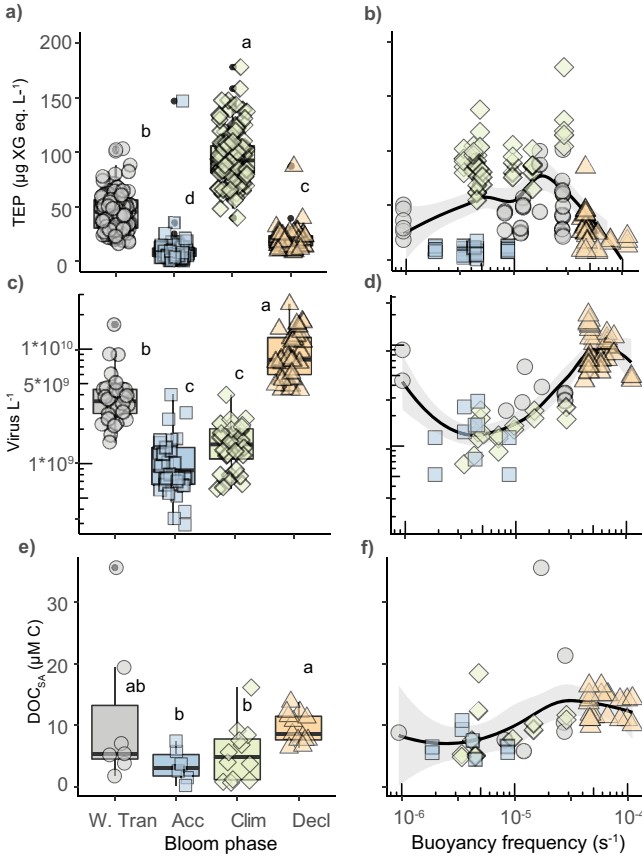

**Fig. 4 Seasonal phases are characterized by distinct extracellular signatures in aggregation potential, virus particle concentration, and DOC accumulation.** Concentrations of **a**, **b** transparent exopolymer particles (TEP: μg Xanthan Gum equivalents L$^{-1}$), **c**, **d** virus-like particles (Virus L$^{-1}$), and **e**, **f** seasonally accumulated dissolved organic carbon (DOC$_{SA}$: μM carbon) within the mixed layer at depths corresponding to 40, 20, or 1% surface irradiance for different seasonal phases (W.Tran = Winter Transition; Acc = Accumulation; Clim = Climax; Decl = Decline). Panels (**b**), (**d**), (**f**) show a subset of samples (corresponding to 5 m sampling depth) plotted as a function of water column stratification (LOESS, shaded area = 95% confidence interval) (buoyancy frequency; s$^{-1}$). Lower buoyancy frequency values to the right of the plot are more stratified. Individual symbols represent biological replicates and are shaped and colored by bloom phase. Box plots represent the median value bounded by the upper and lower quartiles with whiskers representing the median + quartile × 1.5 Different letters denote statistically significant groups ($p < 0.05$, Kruskal−Wallis test with Dunn corrections for multiple comparisons). Intergroup comparisons with more than one letter denote no significant difference between the two groups. Number of biological replicates, by bloom phase (from left to right) = 95, 73, 101, 55 (**a**), 30, 32, 32, 32 (**c**), 7, 6, 12, 12 (**e**). Exact $p$ values can be found in Source Data file.

stability and stratification (Fig. 4d, f)[59]. The dramatic increase in virus production from the Accumulation to Decline phases corroborates high levels of cell death in phytoplankton (Fig. 2c, d, f) and may indicate virus-induced losses as a driver of low phytoplankton accumulation rates observed in highly stratified water columns[60] (Fig. 1c, d).

The high ROS levels in the Climax phase (Fig. 2a, b) may have been linked to virus infection, since positive virus accumulation and elevated virus cell$^{-1}$ started in this phase of the bloom (Supplementary Fig. 5b, e). Our flow-cytometry-based method (Supplementary Fig. 7) could not differentiate between viruses

(and virus-like particles) derived from phytoplankton, heterotrophic bacteria or grazers. It could also not discern if diatoms, which were abundant during the Accumulation and Climax phases[61,62], were producing small ssRNA- or ssDNA-based viruses (20–40 nm diameter)[39], for which flow cytometry-based techniques have not been developed to our knowledge. Instead, they require techniques targeting RNA-dependent RNA polymerase (ssRNA) and replicase (ssDNA) genes, which were not performed in this study. Consequently, our flow cytometry-based data paint a partial picture of how virus production changed with mixed layer depths and across the seasonal North Atlantic bloom.

The enrichment of phytoplankton in the surface layer during Winter Transition (Supplementary Figs. 8a and 9a), absent of bacteria and virus enrichment, indicates a decoupling with these microbes during this phase. As the bloom progressed into the Climax and Decline phase, bacteria (Supplementary Figs. 8b and 9b and Supplementary Table 2) and then viruses and DOC became enriched in the upper 25 m (Supplementary Figs. 8c, d and 9c, d). Viral accumulation in the relatively shallow MLDs of Climax and Decline phases (Supplementary Fig. 5e) may set up an ecosystem where predator−prey encounters are amplified. An ecosystem with high viral lysis can lead to elevated DOC accumulation[50]. In contrast, bloom phases with deeper MLDs (Winter Transition and Accumulation) had significantly lower virus cell$^{-1}$ (Supplementary Fig. 5b) with no surface enrichment of viruses (Supplementary Figs. 8c and 9c); the Accumulation phase also had no surface enrichment of DOC (Supplementary Figs. 8d, 9d).

We related all biomarkers and physical parameters within the mixed layer to each other across bloom phases using cluster analysis with an Optimal Leaf Ordering algorithm (Fig. 5; see 'Methods'). Each variable/biomarker was normalized to its own distribution (log-transformed distribution for biomarkers already transformed in Figs. 2–4) across cruises on a scale of 0 to 1 prior to clustering since the scales and units of individual measurements were fundamentally different. The Accumulation phase had relatively low values of stress- and cell death-associated biomarkers and possessed relatively high accumulation rates, metacaspase activity, and storage lipids. As water columns became more stratified during the Climax phase, both phytoplankton cell concentration and biomass peaked, and oxidative stress, oxidized lipids, and TEP cell$^{-1}$ were most prominent. Stable and shallow mixed layers during the summer/autumn Decline phase brought peak signals in storage lipids, virus cell$^{-1}$, PCD activity, and DOC$_{SA}$. Percent compromised membranes peaked into the Winter Transition phase, but the deeper mixing coincided with generally lower stress markers and lower virus cell$^{-1}$, setting the stage for subsequent positive accumulation (Fig. 5).

These biomarker patterns also held true for populations collected across bloom phases and geographical subregions (Fig. 6). We noted that individual stations within each bloom phase clustered together more than with other stations from similar latitudes and subregions but in different bloom phases (Fig. 6). Indeed, through the lens of these biomarkers, populations within a bloom phase, but deriving from different subregions, resembled each other more than other stations across other bloom phases. This analysis revealed that some biomarkers, such as virus cell$^{-1}$, OxPC, caspase activity, and TEP cell$^{-1}$, clustered more strongly within each bloom phase; other biomarkers such as ROS and metacaspase activity appeared to vary within and between bloom phases (Fig. 6). This implies that some biomarkers represent a seasonal community physiological state that is conserved in mixed layers throughout our study area, while other biomarkers are more sensitive to more subtle bloom timings. Similarly, we noted that certain biomarkers clustered into distinct groups throughout the seasonal bloom. For example,

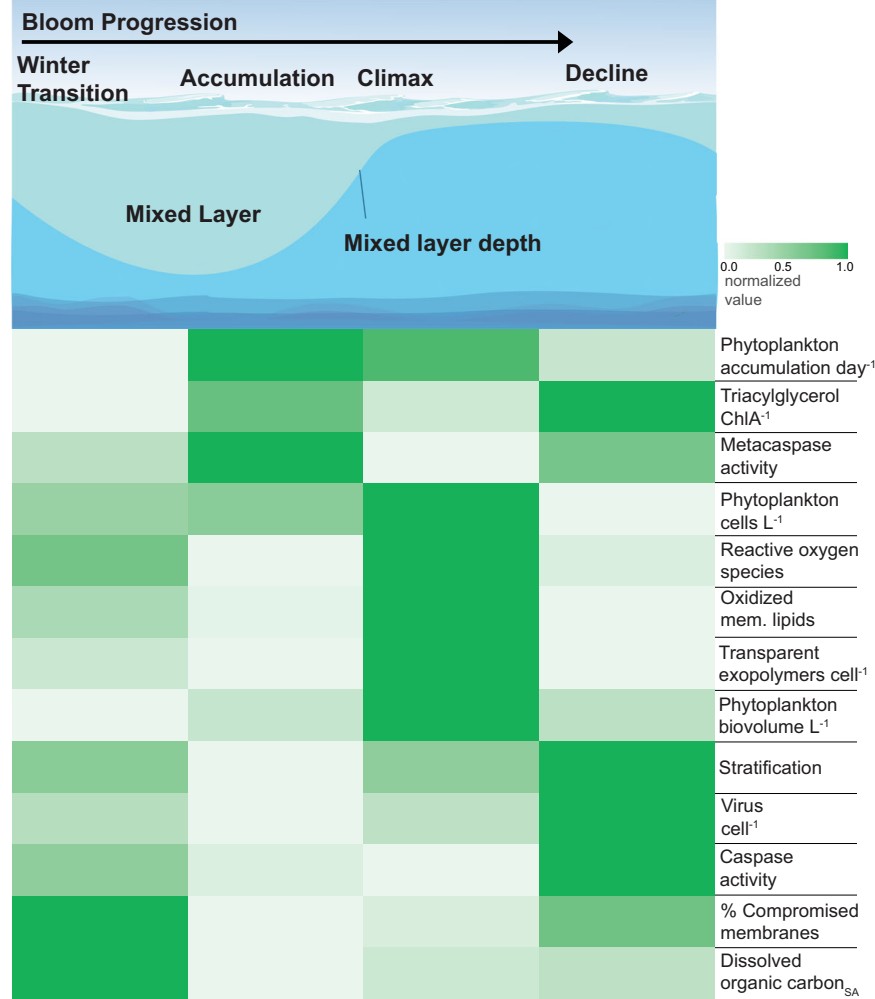

**Fig. 5 Biomarkers of physiological state cluster according to bloom phase and stratification.** (Top) Conceptual illustration of changes in mixed layer depths and water column stratification associated with seasonal bloom phases in the North Atlantic. (Bottom) Heatmap of different biomarkers (organized as rows). Row shading represents the median of each variable throughout a phase, normalized to the entire cycle. Rows are clustered by an optimal leaf algorithm to minimize the distance between normalized distributions of each variable (leaves). Phytoplankton accumulation day$^{-1}$ = cell concentration-based accumulation rate (day$^{-1}$); Triacylglycerol ChlA$^{-1}$ = triacylglycerol concentration (pM) normalized to chlorophyll A; Metacaspase activity = μmol metacaspase substrate cleaved h$^{-1}$ μg protein$^{-1}$; Phytoplankton cells L$^{-1}$ = phytoplankton concentration (cells L$^{-1}$); Reactive oxygen species = log$_{10}$ fold change from unstained phytoplankton population; Oxidized mem. lipids = oxidized phosphatidylcholine normalized to total phosphatidylcholine; Transparent exopolymers cell$^{-1}$ = transparent exopolymer particles (μg XG eq. L$^{-1}$) normalized to bacteria and phytoplankton cell concentrations; Phytoplankton biovolume L$^{-1}$ = phytoplankton biovolume (μL L$^{-1}$); Stratification = measured buoyancy frequency (s$^{-1}$); Virus cell$^{-1}$ = virus concentration normalized to phytoplankton and bacterial cell concentrations; Caspase activity = μmol caspase substrate cleaved h$^{-1}$ μg protein$^{-1}$; % Compromised membranes = % of phytoplankton population with compromised membranes; Dissolved organic carbon$_{SA}$ = seasonally accumulated dissolved organic carbon (μM change from minimum value per bloom phase).

compromised membranes, virus cell$^{-1}$, caspase activity, and DOC$_{SA}$ consistently clustered with degree of stratification (Fig. 6).

Principal component analysis revealed the most pronounced separation in collective physiological markers during Climax and Decline phases (Fig. 7a, b). Recognizing that the biomarker signatures are derived from distinct phytoplankton communities across bloom phases and subregions, our analysis integrated both pigment-based[62] and 16S rDNA-based[61] community analyses to infer taxa identities. Incorporating either community-based analysis, our biomarkers and taxa contributed similarly to covariance in PC1 (Fig. 7c, d). On a bloom-wide scale, stratification positively covaried with cyanobacteria, caspase activity, virus cell$^{-1}$, DOC$_{SA}$, TAG ChlA$^{-1}$ and compromised membranes. These variables negatively covaried with macronutrients, TEP cell$^{-1}$, ROS, OxPC, and diatoms (Fig. 7c, d). Our analysis suggests that diatoms (Climax) and cyanobacteria (Decline) were enriched

in fundamentally different ecosystem/bloom states. It uniquely links these phytoplankton community types to specific cell signatures such as ROS, TEP, cell death, DOC, viruses, and stress lipids.

We further examined biomarker relationships within Climax (Supplementary Fig. 10) and Decline (Supplementary Fig. 11) phases to better understand how biomarkers and community types covaried by bloom phase and stations. TEP cell$^{-1}$, OxPC, and ROS did not covary with nutrient stress (Supplementary Fig. 10c, d) within the Climax phase. TEP cell$^{-1}$ negatively covaried with stratification (Supplementary Fig. 10c, d), indicating that high turbulence[63] may contribute to elevated TEP cell$^{-1}$. ROS did not strongly covary with any biomarker or community type (Supplementary Fig. 10b, d), implying that ROS was a general stressor experienced at all subregions, and possibly linked to UV stress. ROS negatively covaried with nutrients in the

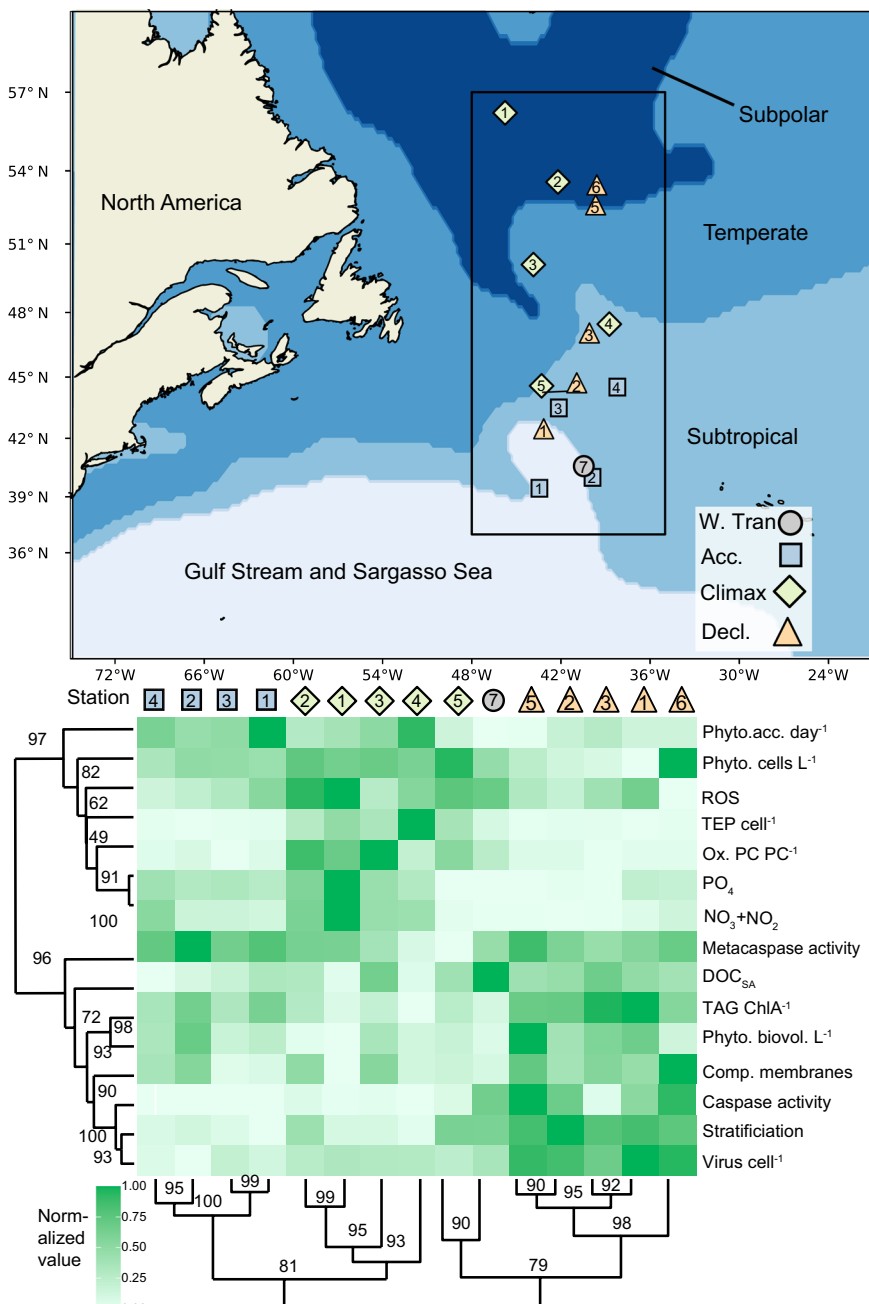

**Fig. 6 Comparison of biomarker signatures between stations and bloom phases in the NAAMES study area.** (Top) Map of stations within subregions and bloom phases used in this analysis. Only stations with in situ values of all of the biomarkers in the heatmap below were included, except for accumulation rates, which were calculated from on-deck incubations. Stations with more than one sampling date were consolidated into a single latitude and longitude for the purpose of this map. Stations are shaped and colored by bloom phase: W. Tran = Winter Transition, Acc. = Accumulation, Climax, Decl. = Decline. (Bottom) Heatmap of normalized values of biomarkers throughout the bloom. The median value at each station, within the mixed layer was normalized to the highest and lowest median values throughout the bloom. Dendrogram clustering was done with the average method, using a correlation distance matrix of normalized values. Bootstrap values ($n = 1000$) are pvclust's AU $p$ value—higher is more significant. Rows and columns were ordered using Dendrogram cluster order. Phytoplankton acc. day$^{-1}$ = cell concentration-based accumulation rate (day$^{-1}$); Phyto. cells L$^{-1}$ = phytoplankton concentration (cells L$^{-1}$); ROS = Reactive oxygen species (log$_{10}$ fold change from unstained); TEP cell$^{-1}$ = transparent exopolymer particle concentration (µg XG eq. L$^{-1}$) normalized to phytoplankton and bacteria concentrations; Ox.PC PC$^{-1}$ = oxidized phosphatidylcholine normalized to total phosphatidylcholine; PO$_4$ = phosphate concentration (µM); NO$_3$ + NO$_2$ = nitrate plus nitrite concentration (µM); Metacaspase activity = µmol metacaspase substrate cleaved h$^{-1}$ µg protein$^{-1}$; DOC$_{SA}$ = seasonally accumulated dissolved organic carbon (µM change from minimum value per bloom phase); TAG ChlA$^{-1}$ = triacylglycerol (µM) normalized to Chlorophyll A peak area/L; Phyto. biovol. L$^{-1}$ = phytoplankton biovolume (µL L$^{-1}$); Comp. membranes = % of population with compromised membranes; Caspase activity = µmol caspase substrate cleaved h$^{-1}$ µg protein$^{-1}$; Stratification = measured water column buoyancy frequency (s$^{-1}$); Virus cell$^{-1}$ = virus concentrations normalized to phytoplankton and bacteria concentrations.

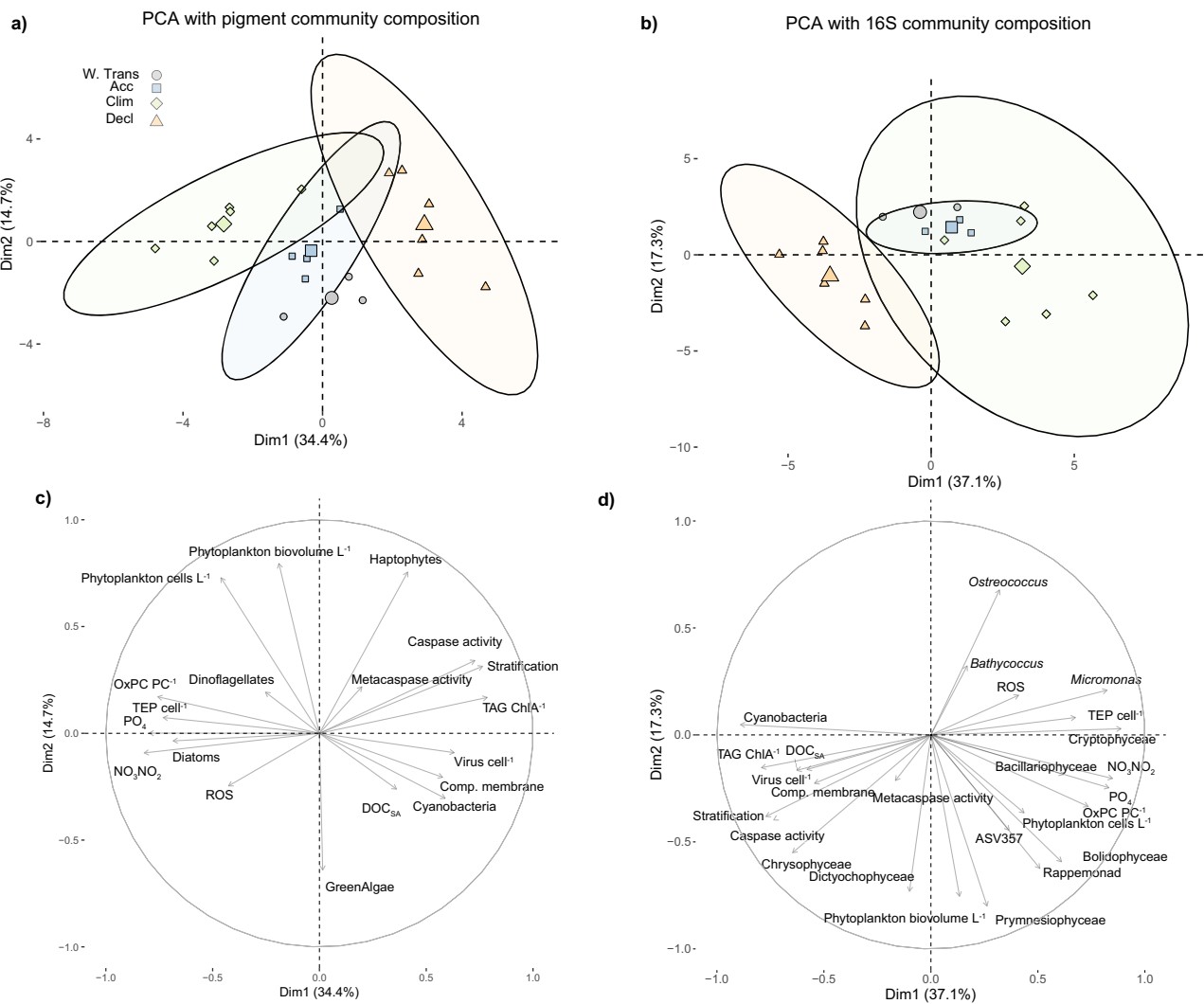

**Fig. 7 Principle component analysis associates bloom phases with distinct physiological states and extracellular signatures.** (Top row) PCA showing interrelationships among measured intra- and extracellular properties for phytoplankton populations across bloom seasons combined with clade relative abundance determined by either **a** Pigment-based community composition[62] or **b** 16S rRNA-based community composition[61]. Symbols are shaped and colored by station. Small symbols represent individual sampling events during each bloom phase; large symbols represent the average value for each respective bloom phase. Percent variability in each dimension is the amount of variance explained in the dataset. Ellipses represent the 95% concentration estimate. W.Tran = Winter Transition; Acc = Accumulation; Clim = Climax; Decl = Decline. (Bottom row) PCA scores and vectors for measured intra- and extracellular properties with respect to **c** Pigment-based community composition and **d** 16S rRNA-based community composition. Longer arrows represent a stronger correlation with each respective dimension. Phytoplankton cells $L^{-1}$ = phytoplankton cell concentration; ROS = reactive oxygen species in phytoplankton ($\log_{10}$ fold change in fluorescence from unstained); Comp. membrane = % of phytoplankton population with compromised membranes; TEP cell$^{-1}$ = transparent exopolymer particle concentration (μg XG eq. $L^{-1}$) normalized to bacteria and phytoplankton concentration; Virus cell$^{-1}$ = virus concentration normalized to bacteria and phytoplankton concentrations; OxPC PC$^{-1}$ = oxidized phosphatidylcholine (peak area) normalized to total phosphatidylcholine; TAG ChlA$^{-1}$ = triacylglycerol (pM) normalized to ChlA (peak area/L); Caspase activity = caspase activity (μmol caspase substrate cleaved h$^{-1}$ μg protein$^{-1}$); Metacaspase activity = metacaspase activity (μmol metacaspase substrate cleaved h$^{-1}$ μg protein$^{-1}$); DOC$_{SA}$ = seasonally accumulated dissolved organic carbon; Phytoplankton biovolume $L^{-1}$ = phytoplankton μl $L^{-1}$; Stratification = stratification of the upper 300 m, expressed as buoyancy frequency (s$^{-1}$). Biomarkers are plotted according to their co-variability with phytoplankton groups defined by pigments (i.e., Green Algae, Cyanobacteria, Diatoms, Dinoflagellates, Haptophytes) or 16S (i.e., Cyanobacteria, Chrysophyceae, Dictyochophyceae, Prymnesiophyceae, Rappemonad, Bolidophyceae, ASV357, Bacillariophyceae, Cryptophyceaea, Micromonas, Bathycoccus, Ostreococcus).

Decline phase (Supplementary Fig. 11b, d) suggesting that here nutrient limitation may have contributed to higher ROS. Overall, metacaspase activity was not strongly covarying with any one biomarker or group in our analyses (Fig. 7c, d and Supplementary Figs. 10, 11), which is supportive of their reported involvement in a variety of healthy or stress-related functions unique to different taxa[7,46].

Integrating community composition revealed some associations between biomarkers and community type. TEP cell$^{-1}$

notably clustered with communities in which diatoms were in higher relative abundance by either pigment- or 16S rRNA-based community composition analysis (Fig. 7c, d and Supplementary Fig. 10c). Some diatom species are known to have significantly higher TEP cell$^{-1}$ compared to other phytoplankton[49], so they may have been a source of TEP. 16S-based community composition showed that *Micromonas* and Cryptophyceae also positively covaried with TEP cell$^{-1}$, implying that a community type, rather than one group of phytoplankton, was associated

with high TEP cell$^{-1}$. Although cyanobacteria positively covaried with compromised membranes, TAGs, caspase activity and viruses in the highly stratified waters of the Decline phase (Fig. 7 and Supplementary Fig. 11), they were likely not the source of these biomarkers for several reasons. First, cyanobacteria did not positively covary with compromised membranes and caspase activity; likewise, they only weakly positively covaried with virus cell$^{-1}$ (Supplementary Fig. 11c, d). Second, our compromised membrane detection procedure did not include cyanobacteria due to cell size discrimination (Supplementary Table 1). Third, TAGs are not known to be a major component of marine cyanobacteria biomass[32]. They are more likely the product of eukaryotic phytoplankton[42,43] or zooplankton metabolism[34] in a nutrient limited environment. Fourth, caspase/metacaspase orthologs have not been identified in the genomes of unicellular cyanobacteria, such as *Prochlorococcus* and *Synechococcus*[29]. Lastly, lytic virus production and cell death would mechanistically reduce host cell concentrations; we instead observed that higher total virus concentrations positively covaried with cyanobacteria populations. Additional size fractions, high speed cell-sorting, and/or single-cell analyses, combined with nucleic acid-based techniques would provide finer resolution on which taxa experienced specific physiological changes and paint a more comprehensive picture of virus infection dynamics throughout the North Atlantic bloom.

**Physiological responses to mixing and stratification on shorter time scales.** We further explored our biomarker data between stations occupied during NAAMES as case studies for community physiological responses to different mixing scenarios. This allowed us to discern biomarker patterns over shorter temporal and spatial scales. Climax Station 4 provided an opportunity to assess changes in stress and death biomarkers within the same phytoplankton population in a water column, which transitioned from a deep to a shallow MLD. Upon occupation of the station, we observed a deep MLD (>230 m), which shallowed to <20 m during our 2-day occupation (Fig. 8). This physical transition did not significantly alter[61] community composition. We determined the in situ population physiology over these 2 days and for an additional 5 days using on-deck incubations.

The respective physiological states of phytoplankton communities residing in the upper 25 m of deep (Fig. 8, pink symbol) and shallow (Fig. 8, yellow symbol) mixed layers showed key differences. Re-stratification of Climax Station 4 stimulated both phytoplankton and viral accumulation. Increased phytoplankton accumulation is likely due to the increased daily availability of light and reduced grazer presence[64], while increased virus L$^{-1}$ may be indicative of a subset of phytoplankton or bacteria lysing/releasing viruses into the surrounding water during periods of strong stratification. Phytoplankton populations residing within the deeper mixed layer also had notable physiological differences. These populations initially had higher compromised membranes (~20%, Supplementary Fig. 12m), which decreased in subsequent days of occupation (Fig. 8). We point out that this level of compromised membranes is close to the values seen in the relatively healthy Accumulation phase (Fig. 2). Populations residing in the deep mixed layer also had lower TEP cell$^{-1}$ compared to the communities in more stratified surface waters (Fig. 8). For comparison, the Accumulation phase, which also harbored deeply mixed populations (Fig. 1b), had lower TEP cell$^{-1}$ and virus cell$^{-1}$ relative to this phase (Supplementary Fig. 5a, b). The increased turbulence during the episodic deep mixing event at Climax Station 4 may have facilitated the abiotic aggregation[63] of dissolved TEP precursors making them large enough for retention on the 0.45 μm pore-size filters (see 'Methods'). The relatively high TEP cell$^{-1}$ could have also been derived from production by

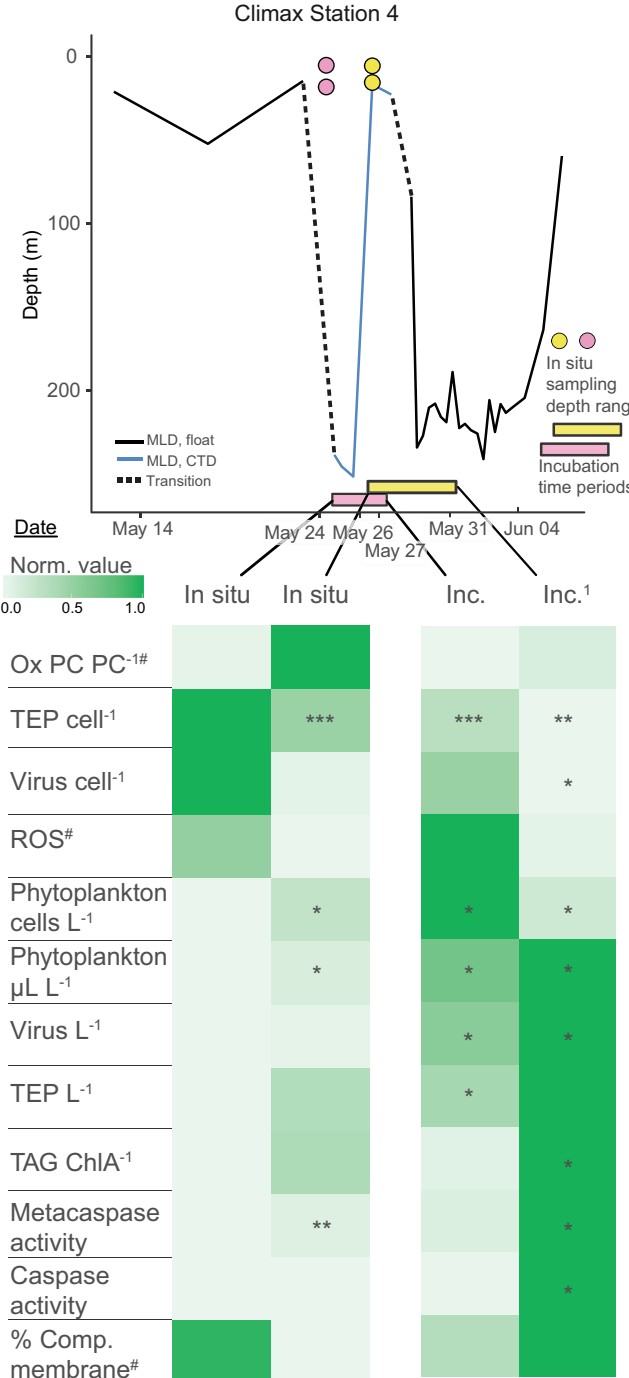

colonizing bacteria on cellular detritus[51,65] brought to the surface by active mixing. Even though TEP cell$^{-1}$ decreased, bulk (non-cell normalized) TEP L$^{-1}$ increased as the mixed layer shallowed (Fig. 8). Increased TEP L$^{-1}$ is consistent with general increases in phytoplankton biomass[49,66–68], or higher daily irradiance[67], both of which occurred during this period.

We next compared in situ phytoplankton at Climax Station 4 to the community at Climax Station 1. These stations were in the same bloom phase with similar phytoplankton concentrations (8.0–8.3 × 10$^6$ cells L$^{-1}$) but had different mixing histories. Climax Station 1 had a consistently shallower MLD (ranging from 42 to 56 m) two weeks prior to and during our occupation, compared to the recently mixed and stratified Climax Station 4 (from 233 to 17 m). Given the general increase in stress and death

**Fig. 8 Physiological state of phytoplankton transitioning from a deep to a shallow mixed layer.** (Top) Mixed layer depth at Climax Station 4 in relation to sampled phytoplankton communities. Solid black line represents mixed layer depth (MLD) pre- and post-occupation provided by drifting BioArgo floats (solid black line). A dashed line represents the transition from the float-measured MLD to ship-measured MLD (solid blue line), since sampling was done near and not directly at the float. Water sampled from the first and third days of occupation are indicated by pink and yellow symbols, respectively. The y axis positions of colored circles represent sampling depths. The x axis position of colored circles represents the date of sampling. Colored bars on the x axis correspond to incubation times of sampled communities (colored circles were the source water). (Bottom) Comparison of intra- and extracellular biomarkers from phytoplankton associated with in situ and incubation samples from above. Color shading in each row represents the normalized distribution of each parameter over the 8-day transition from a deeply mixed to a shallow mixed layer. Row order is clustered by optimal leaf algorithm. Data for incubations derive from the same in situ water collected on days 1 and 3, respectively, and incubated on deck at in situ light and temperature (see 'Methods'). Asterisks indicate significant differences from day 1 via Kruskal−Wallis test (*$p < 0.05$, **$p < 0.01$, ***$p < 0.001$). Ox PC $PC^{-1}$ = oxidized phosphatidylcholine normalized to total PC; TEP $cell^{-1}$ = transparent exopolymer particles normalized to phytoplankton and bacterial cell concentrations; Virus $cell^{-1}$ = virus concentration normalized to phytoplankton and bacterial cell concentrations; ROS = reactive oxygen species; Phytoplankton cells $L^{-1}$ = phytoplankton cell concentration; Phytoplankton µl $L^{-1}$ = phytoplankton biovolume; Virus $L^{-1}$ = virus concentration; TEP $L^{-1}$ = transparent exopolymer concentration; TAG $ChlA^{-1}$ = triacylglycerol concentration (pM) normalized to chlorophyll A peak area/L; Metacaspase activity = µmol metacaspase substrate cleaved, µg $protein^{-1} h^{-1}$; Caspase activity = µmol caspase substrate cleaved, µg $protein^{-1} h^{-1}$; % Comp. membrane = % of population with compromised membranes. [1]Incubation samples from depths lower than 5 m were lost due to a storm knocking the incubation tanks off the deck. #$n < 3$ samples for at least 1 day for these parameters, due to loss of samples in transit or lack of sampling on day 1. See Supplementary Fig. 12 for raw parameter data from this station. Exact p values can be found in Source Data file.

markers for these populations after a five day incubation (Fig. 8), we expected cell populations at Climax Station 1 to resemble the communities at Climax Station 4 after stratification/incubation. Accordingly, Climax Station 1 had higher levels of intracellular oxidative stress, virus $L^{-1}$, metacaspase, and caspase activity (Fig. 9) than freshly mixed populations from Climax Station 4. This implies that populations in more stable water columns are transitioning to ROS-induced stress, increased PCD, and viral infection. Based on pigment composition, both days of Climax Station 4 were classified as diatom community types. In contrast, Climax Station 1 had a higher relative abundance of diatoms on the first day of occupation and dinoflagellates on the second day of occupation. Based on 16S rRNA analysis, the most abundant group at Climax Station 4 for both days of occupation was *Ostreococcus* (37 and 44% relative abundance, R.A.), while Climax Station 1 contained mainly *Micromonas* (47% R.A.). Collectively, these data may indicate succession to dinoflagellate or chlorophyte populations as the water column became more stratified during this bloom phase.

Lastly, we compared two stations in the Decline phase that had relatively stable, shallow mixed layers prior to and during occupation, but were characterized by relatively high (Decline Station 6: $2 \times 10^7$ cells $L^{-1}$) or low (Decline Station 2: $3 \times 10^6$ cells $L^{-1}$) phytoplankton concentrations. Decline Station 2 had higher oxidative stress (ROS, OxPC) and TAGs (Fig. 9). These data are consistent with nutrient stress[8,12] and virus infection[13], both of

which could contribute to lower phytoplankton biomass. Since Decline Station 2 had lower virus $cell^{-1}$ and virus $L^{-1}$ than Decline Station 6, (Fig. 9 and Supplementary Fig. 13), the elevated TAG and ROS at this station more likely came from nutrient limitation than viral infection. Decline Station 2 was located in the subtropical north Atlantic, which was relatively more nitrate limited (Fig. 6), which agrees with modeling predictions of these subregions[69]. Decline Station 6, on the other hand, was in higher latitude subpolar waters and did not appear to be macronutrient limited (Fig. 6). These differences in nutrient concentration between these subregions in Decline phase may have shaped community composition and its impact on the aforementioned biomarkers (TAG, ROS, OxPC). Pigment analysis showed that Decline Station 6 and Decline Station 2 were respectively haptophyte and cyanobacteria community types for both sampling dates. Cyanobacteria were still the most prominent group in Decline Station 6 based on 16S rRNA analysis but they were less dominant than at Decline Station 2 (30, 32% vs. 86% R.A., respectively). Decline Station 6 contained more Bacillariophyceae (diatoms) (13% vs. 0.1% R.A.) and Prymnesiophyceae (haptophyte group) (~13% vs. 1% R.A.).

Resident populations in Decline Station 6 had significantly higher TEP $cell^{-1}$, compromised membranes, and caspase activity compared to Decline Station 2 (Fig. 9), consistent with active viral infection and lysis[29,50]. Compromised membranes and virus $cell^{-1}$ both increased during at this station (Supplementary Fig 14), even though total phytoplankton concentration did not decrease, even 6 days after initial occupation (Supplementary Fig. 14). This finding further confirms that Decline Station 6 was not nutrient stressed and maintained a high biomass in the presence of top-down processes such as grazers or viruses. We note that viruses started accumulating in the Climax phase (Supplementary Fig. 5e) and resulted in universally high virus $cell^{-1}$ in the Decline phase (Supplementary Fig. 5b), regardless of latitude, sub-regional water type or macronutrient variability (Fig. 6). Our analysis suggests that this high virus $cell^{-1}$ is a persistent ecosystem pressure for phytoplankton, bacteria, or possibly grazers living in stratified water columns in early autumn in the Western North Atlantic.

In summary, our findings provide evidence that seasonal water column stratification is correlated with intracellular stress, PCD, and viral predation in phytoplankton communities. These cellular loss processes are variable and follow water column stratification, which effectively governs biomass accumulation over the annual phytoplankton bloom cycle in the North Atlantic. High phytoplankton accumulation occurs with an absence of physiological stress markers in the deeply mixed early spring. Phytoplankton cells continue to accumulate into the Climax phase as the mixed layer shallows in the late spring. Intracellular ROS, oxidized lipids, and TEP $cell^{-1}$—all signatures of oxidative stress—became elevated in this phase, signaling a transition towards loss in the Decline phase. The lowest accumulation rates of phytoplankton biomass throughout the bloom occurred within physically stable and shallow mixed layers of the Decline phase. While nutrient availability varied with sub-region, communities in this phase had the highest compromised membranes, cell death protease activity, virus $cell^{-1}$, $DOC_{SA}$, and TAG $ChlA^{-1}$ across all stations and subregions. These Decline phase biomarkers have been extensively linked to cell death and late stages of viral infection in diverse model systems in culture[29,43,50,70]. Deepening of the mixed layer during Winter Transition increased nutrient concentrations and diminished both nutrient stress biomarkers and virus $cell^{-1}$, setting the stage for renewed phytoplankton accumulation. Community stress responses were also discernable on a shorter temporal scale between locations with different mixed layer histories and in response to a mixed layer shallowing

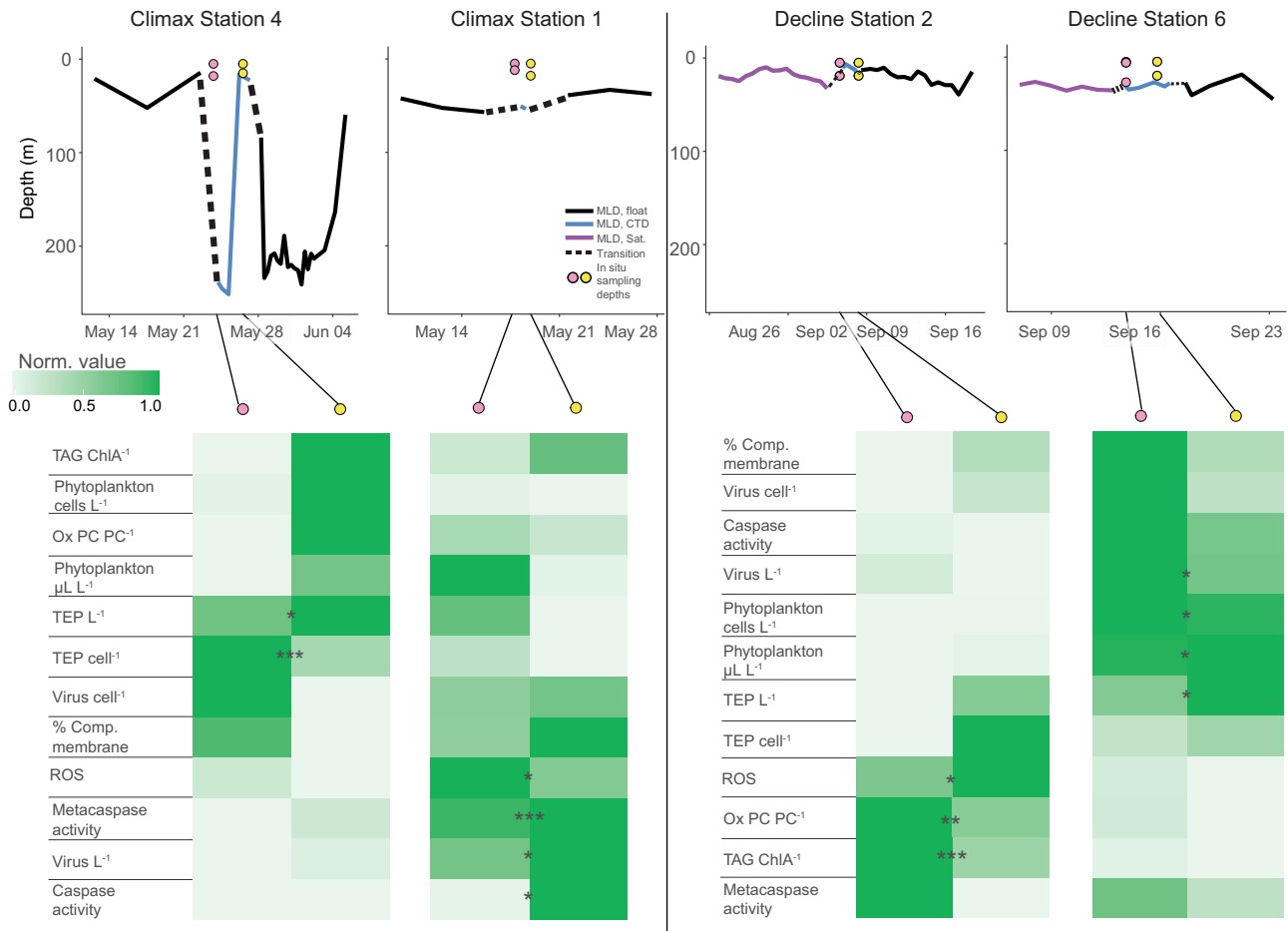

**Fig. 9 Inter-station comparisons of phytoplankton physiological state in water columns with different mixing depths and biomass loads.** (Top) Mixed layer depth dynamics at a subset of Climax and Decline stations in relation to sampled phytoplankton communities. Climax Station 4 transitioned from a deeply mixed to a shallow water column, while Climax Station 1 had more consistent MLDs during station occupation. More stable and shallow MLDs were observed during Decline phase, but phytoplankton communities had different relative biomass levels (Decline Station 6, high; Decline Station 2, low). Pre- and post-occupation mixed layer depths (MLD) are provided by drifting BioArgo floats (solid black line) or derived from satellite measurements (purple line), since BioArgo floats were not available for the Decline phase. A dashed line represents the transition from the float-measured MLD to ship-measured MLD (solid blue line), since sampling was done near and not directly at the float. y axis positions of colored circles represent sampling depths of in situ populations. x axis position of colored circles represents the date of sampling. Dates and data corresponding to colored symbols are used in the comparison table below. (Bottom) Comparison of intra- and extracellular biomarkers for phytoplankton associated with in situ samples from above. Pink symbols = first day of occupation. Yellow symbols = subsequent day of occupation. Rows represent a two-day time course of each measured biomarker variable. Color shading represents the normalized distribution of each parameter within each seasonal phase to better illustrate relative responses to MLD dynamics. Row order is clustered by optimal leaf algorithm. Asterisks indicate significant differences between stations, via Kruskal−Wallis test (*$p < 0.05$, **$p < 0.01$, ***$p < 0.001$). Biomarker definitions are the same as in Fig. 8. OxPC PC$^{-1}$ also used additional data from the 1% light depth due to some samples being lost in transit. See Supplementary Fig. 13 for raw parameter data from these stations. Exact p values be found in Source Data file.

event. The stress patterns we observed on different time scales provide insight to a previously unappreciated coupling of physics and phytoplankton physiology that can help inform our understanding of the underlying processes that govern phytoplankton bloom dynamics across the western North Atlantic.

## Methods

**Fieldwork.** The North Atlantic Aerosol and Marine Ecosystem Study (NAAMES) consisted of four independent cruises in the northwest Atlantic aboard the R/V Atlantis, each traversing a ~2000 nautical mile transect[22] (see Fig. 1a for locations sampled). Each cruise was performed at a different time of the year corresponding to different seasonal phases of the annual phytoplankton biomass cycle in the North Atlantic [NAAMES1, 6 November–1 December 2015 (Winter Transition); NAAMES2, 11 May–5 June 2016 (Climax); NAAMES3, 30 August–24 September 2017 (Decline); NAAMES4, 20 March–13 April 2018 (Accumulation)]. A suite of biological, chemical, physical, and optical properties were performed during each cruise to contextualize each phase of the annual cycle[22]. Sampling was done in international waters.

Discrete water samples were collected within 4–7 station locations during each phase using 10-L Niskin bottles mounted on a 24-position rosette equipped with a Seabird SBE conductivity-temperature-depth (CTD) profiler. Water was collected in two different ways for analyses. To obtain in situ biomarker data, samples were collected using a CTD cast 2–4 h before sunrise ('pre-dawn') at depths corresponding to 40, 20, and 1% surface light levels. Water from the same depths as in situ samples was also emptied directly into acid-washed, 9 L bottles and placed in on-deck incubators, which were continuously circulated with surface water to maintain ambient temperatures and covered with neutral density screening to simulate 40, 20, and 1% surface irradiance.

Diagnostic staining for reactive oxygen stress and compromised membranes was performed within 2 h of sampling. Water was also filtered onto membrane filters to collect host cell biomass for lipid and protein enzymatic rates (see below for details). All filters were snap-frozen in liquid nitrogen and stored at −80 °C until processed. Seawater samples were kept in the dark at in situ temperature prior to processing in the lab on the R/V Atlantis.

Water was also collected at six depths (mostly within the upper mixed layer and extending down to 150 m) during 'core casts' conducted later the same morning and sampled for phytoplankton, virus, TEP and DOC concentrations. All filters were snap-frozen in liquid nitrogen and stored at −80 °C until processed.

**Enumeration of phytoplankton, bacterioplankton and viruses.** Analytical flow cytometry of water collected during CTD casts was used to enumerate phytoplankton (1–20 μm cell diameter), bacteria, and viruses. Phytoplankton concentration was determined with a BD Accuri C6 (flow rate of 14 μL per minute and for 3–7 min per replicate, software version 1.0.264.21) using a 488 nm laser to excite cells and an 80,000 FSC-H threshold (forward scatter height). Polystyrene beads (Spherotech; 2, 3.4, 5.1, 7.4, 10, and 14.3 μm diameters) and unialgal cultures (Phaeocystis globosa provided by Lee-Karp Boss at University of Maine and Emiliania huxleyi from the Bidle lab at Rutgers) were used to confirm phytoplankton gates.

Virus and bacteria concentrations were determined after each cruise from fixed and frozen samples. At each in situ and incubation sampling time point, glutaraldehyde was added to an aliquot of seawater for a final concentration of 0.5%. Samples were incubated for 10–30 min at 4 °C and then flash frozen in liquid nitrogen and stored at −20 °C. Samples were diluted 1/50 into 0.22 μm filtered TE buffer pH 8 with SYBR Gold (Thermo Fischer) added to 1× concentration, then heated at 80 °C for 10 min in the dark. After cooling in the dark, samples were counted on the BD Influx Mariner using 520 nm as the threshold wavelength. SYBR Gold-stained, TE Buffer controls were used to determine background noise. Size calibration beads (Spherotech, 0.2, 0.5 μm diameter) and model Emiliania huxleyi virus (EhV 207) lysates were used to calibrate virus sizes via forward scatter. Sample events between ~50 and 200 nm were determined to be viruses, while between 200 nm and 0.5 μm were counted as bacteria (Supplementary Fig. 4).

**Calculation of phytoplankton biovolume.** Phytoplankton biovolume was calculated using a standard curve derived from polystyrene beads run with 80,000 FSC-H threshold on the BD Accuri (Spherotech; 2, 3.4, 5.1, 7.4, 10, and 14.3 μm diameters). Median FSC-H values from each phytoplankton cell event (FSC-H > 80,000, FL3-H > 3500) were converted to diameter using the following linear regression, assuming a sphere to calculate biovolume:

$$\text{Equivalent spherical diameter}(\mu m) = (FSC - H) \times 5 \times 10^{-6} + 0.8367 \quad (1)$$

**Calculation of phytoplankton accumulation rates.** Phytoplankton accumulation was calculated using the following equation:

$$[\text{Phytoplankton accumulation day}]^{-1} = (\ln(C_2/C_1))/(T_2 - T_1) \quad (2)$$

where $C_1$ and $C_2$ are the respective cell concentrations at an initial time point ($T_1$) and after time ($T_2$) in days. Incubation accumulation rates were determined from the phytoplankton concentrations in 9 L bottle incubations over a 48–72 h incubation period, with the initial sample coming directly from the CTD Niskin bottle.

In situ accumulation rates were determined by sampling 5 m water from the clean intake on the R/V Atlantis at time intervals of 1–8 h. In situ phytoplankton accumulation rates shown in Fig. 1c were measured with an Influx Mariner flow cytometer, with phytoplankton defined as the sum of cyanobacteria, picophytoplankton, and nanophytoplankton. All other in situ- and incubation-based phytoplankton accumulation rates were determined using the BD Accuri and the size criteria above (1–20 μm cell diameter).

**Transparent exopolymer particles.** 150 mL of seawater was filtered onto a 25 mm, 0.45 μm pore-size PC filter in triplicate, stained with 0.02% Alcian Blue solution (acidified with acetic acid[71]), and rinsed 3× with 1 mL MilliQ water. Stained filters were flash frozen in liquid nitrogen and kept at −80 °C until analyzed. Filters were thawed at room temperature, placed in chemically resistant, plastic, BRAND UV cuvettes, and incubated with 2 mL of 80% sulfuric acid for 2 h, shaking gently. Filters were removed from sulfuric acid solution and the adsorption of the supernatant was measured at 787 nm on a Molecular Devices SpectraMax M3 spectrophotometer (SoftMax Pro version 6.3 software). Calibration curves were made with xanthan gum diluted in water and stained with Alcian Blue solution[72]. TEP cell⁻¹ was calculated by dividing TEP L⁻¹ by the sum of phytoplankton cells (1–20 μm diameter) and bacteria cell concentrations.

**Calculation of TEP and virus accumulation rates.** TEP and virus accumulation were calculated using the following equation:

$$\text{TEP or virus accumulation day}^{-1} = (C_2 - C_1)/(T_2 - T_1) \quad (3)$$

where C1 and C2 are the respective viral or TEP concentrations at an initial time point (T1) and after time (T2) in days. All accumulation rates were determined from 9 L bottle incubations over a 48–72 h incubation period, with initial sample coming directly from the CTD Niskin bottle.

**Staining for cellular reactive oxygen species and compromised membranes.** Diagnostic staining coupled with flow cytometry was used to assess phytoplankton physiological states. Levels of intracellular reactive oxygen species (ROS) were determined by staining cells with CM-H2DCFDA (Thermo Fischer) at a final concentration of 5 μM and incubating in the dark for 60 min. Median fluorescence values (Ex. 488, Em. 533/30 nm) of CM-H2DCFDA-stained samples were divided by the median fluorescence values for unstained controls to account for differences in cell size. CM-H2DCFDA staining of hydrogen peroxide-treated samples (10 μM final concentration) was used as a positive control and to verify the efficacy of ROS staining.

Levels of compromised membranes and cell viability (live versus dead) were assessed by staining cells with SYTOX Green (Thermo Fisher) at a final concentration of 1 μM and incubating in the dark for 10 min. Glutaraldehyde-fixed (0.5% final concentration) cells from the same water were stained with SYTOX Green served as a positive control to determine the maximum fluorescence level (Ex. 488, Em. 533/30 nm) of dead cells in that population, empirically setting thresholds for positively stained cells at each station. Cells from sampling locations were also heat-treated to 80 °C for 10 min and stained with SYTOX as an additional positive control.

**Particulate lipid analysis.** Samples for total lipids analysis were collected onto 0.2 μm pore-size Durapore membrane filters (GVWP-type; Millipore) and immediately frozen in liquid nitrogen for storage and subsequent transport to the laboratory. Samples were extracted by using a modified Bligh and Dyer extraction[73]; an internal standard of 2,4-dinitrophenyl-modified phosphatidylethanolamine (DNP-PE) was added during extraction to account for variations in extraction recovery[73]. Total lipid extracts were analyzed by high performance liquid chromatography/electrospray-ionization high-resolution accurate-mass mass spectrometry (HPLC/ESI HRAM MS) using an Agilent 1200 HPLC coupled to a Thermo Scientific Q Exactive HRAM MS.80 Thermo Scientific Xcalibur v. 3.1 with LOBSTAHS data analysis package (version 1.18.1)[74], along with characteristic retention time and MS2 fragmentation spectra were used to identify lipid classes (Supplementary Fig. 4); samples were quantified relative to external standard calibration curves[35,75]. Reported OxPC PC⁻¹ levels refer to the ratio of oxidized forms of PC 40:10, PC42:11, and PC44:11 to the sum of oxidized and non-oxidized forms of PC 40:10, PC 42:11, and PC 44:11, which were identified and quantified[15].

**Caspase and metacaspase activities.** One liter of seawater from each 40, 20 and 1% surface irradiance depths was collected and filtered in triplicate by vacuum onto 47 mm, 0.8 μm pore-size PC membrane. Filters were snap-frozen in liquid nitrogen and kept frozen until processing back in the lab. Caspase and metacaspase activities were determined by extracting proteins from half of a frozen filter in caspase reaction buffer (50 mM HEPES, 50 mM NaCl, 0.1% CHAPS, 10 mM EDTA, 5% glycerol, 10 mM dithiothreitol, pH 7.2). Frozen filters were kept on dry ice and cut using sterile scalpel. Cells were lysed using a Misonix probe sonicator on power setting 2 with three alternating 30 s pulse and rest cycles (all on ice), centrifuged (10,000 × g, 10 min, 4 °C), and the resultant supernatant collected and flash frozen. Extracts were incubated in 96-well plates with 50 μM IETD-AFC (caspase[47]) or 50 μM VRPR-AMC (metacaspase[76]). Kinetic readings of fluorescence (Ex. 400 nm, Em. 505 nm) were taken every 5 min for several hours on a Molecular Devices SpectraMax M3 spectrophotometer. Cleavage rates were calculated from the slopes over the first 30 min (VRPR-AMC) and 60 min (IETD-AFC). Cell extracts without added substrate and substrates without cell extract both served as negative controls. Samples with slopes lower than the control were considered as a slope of zero. AFC and AMC standard curves (in buffer) were used to convert relative fluorescence values to moles of substrate cleaved. Protein concentrations in cell extracts were determined via Pierce protein assay (Thermo Scientific) and used to calculate specific activities (activity per unit protein).

**Dissolved organic carbon.** DOC concentrations (μmol carbon L⁻¹) were determined from replicate samples drawn from 15 depths from the surface down to 1500 m at each station (nominally 5, 10, 25, 50, 75, 100, 150, 200, 300, 400, 500, 750, 1000, 1250, and 1500 m). Samples were gravity filtered directly from the Niskin bottles through pre-combusted (4 h at 450 °C) GF/F filters and into pre-combusted (4 h at 450 °C) 40 mL borosilicate glass vials. Immediately after collection, samples were acidified to a pH of <3 with the addition of 4 N HCl. Samples were stored at ~14 °C in an environmental chamber free of volatile organics until analysis at the University of California, Santa Barbara. DOC concentrations were measured by high temperature combustion using Shimadzu TOC-V or TOC-L analyzers[77]. Each analytical run was calibrated using glucose solutions of 25–100 μmol C L⁻¹ in low carbon blank water. Precision for DOC analysis has a CV of ~2% or ~1 μmol L⁻¹ for these data. Surface and deep seawater references (sourced from the Santa Barbara Channel), calibrated with DOC consensus reference material (CRM) provided by D. Hansell (University of Miami)[78] were run every 6–8 samples to assess analytical run quality and quality control of DOC data[77].

**Seasonally accumulated DOC.** Seasonally accumulated dissolved organic carbon (DOC$_{SA}$, μmol C L⁻¹) within the euphotic zone was calculated as the difference between locally measured DOC concentration profile (at each station) and the DOC concentrations estimated for periods of maximal deep mixing at each station.

Deeply mixed DOC concentrations ($\mu$mol C L$^{-1}$) for each station within the NAAMES study region were estimated by redistributing (i.e. depth-normalized integration) stratified 'Decline' phase DOC profiles over their corresponding local maximum MLDs, determined from ARGO float observations[21,79].

**16S rRNA-based community analysis.** Four liters of sub-surface seawater (5 m) was filtered through a 0.22 $\mu$m pore-size Sterivex filter cartridge (polyethersulfone membrane, Millipore). One milliliter of sucrose lysis buffer was added to each cartridge and filters were stored at $-80$ °C until further processing. DNA was extracted from the filters using a phenol:chloroform protocol. The hypervariable V1$-$V2 region of the 16S rRNA gene was amplified with the 27F and 338 RPL primers attached to Illumina overhang adapters (Illumina Inc.). Libraries for each reaction product were constructed by attaching dual indices with the Nextera XT Index Kit (Illumina Inc.) using a second PCR amplification (following manufacturer's conditions). All PCR reactions were purified using AMPure XP beads (Beckman Coulter, Brea, CA, USA) and pooled by bloom phase. Each pooled library was sequenced using the Illumina MiSeq platform (reagent kit v.2; 2 × 250 PE; Illumina Inc.). Previously analyzed taxonomic identifications and relative abundance were used[61], leaving low abundance groups out and combining all cyanobacteria into a single group.

**HPLC pigment-based community analysis.** Between 1 and 2 L of whole seawater samples from Niskin bottles and from the flow-through system were collected via filtration onto 25 mm GF/F filters (Whatman®). Filters were immediately stored in liquid nitrogen after filtration and were kept in liquid nitrogen or at $-80$ °C until sample analysis. High performance liquid chromatography was performed at the NASA Goddard Space Flight Center, following predetermined quality assurance and quality control protocols[80,81]. Empirical Orthogonal Function analysis (using the pca functions) and network-based community detection analysis (using the modularity_und.m function) were performed in MATLAB R2020a. The mean ratios of accessory pigments to total ChlA in each community were then used to determine the taxonomic significance of the community[62].

**Contribution of virus particles and bacterial cells to DOC pool.** The relative contribution of carbon from viral particles and bacteria cells was calculated using conservative estimates of 0.2 fg per viral particle[57], and 12.4 fg per bacterial cell[58], along with flow cytometry-based concentration measurements. The method used to segregate DOC from total organic carbon (~0.7 $\mu$m pore-size GFF filtrates) includes ~60% of the bacteria and all the viruses present. The corrected levels of DOC and DOC$_{SA}$ were indistinguishable within the CV of ~1 $\mu$mol L$^{-1}$ confirming that elevated virus and bacterial particles themselves had negligible effects on the measured DOC concentrations (Supplementary Fig. 6).

**Clustering and heatmap construction of biomarkers.** Intra- and extracellular biomarkers for phytoplankton were clustered using an Optimal leaf ordering (OLO) algorithm in the heatmaply package version 1.1.0 in R, which performs a dendrogram analysis based on the optimized Hamiltonian path length to find which variables covaried together. Data were pretreated with the normalize function, which subtracts the minimum value and divides by the maximum value to bring each distribution to a scale of 0 to 1.

**Degree of stratification and mixed layer depths.** The stratification index or degree of stratification is based on set threshold levels averaged over the upper 300 m water column. Brunt-Väisälä or buoyancy frequency (N$^2$; expressed in s$^{-1}$), was calculated as:

$$N^2 = (-g)/\rho \, \partial \rho / \partial z \qquad (4)$$

where $z$ is depth (m), $\rho$ is the potential density of seawater (kg m$^{-3}$) and $g$ is the gravitational acceleration (9.8 m s$^{-2}$). Satellite-derived mixed layer depths were obtained by 2D interpolation of ~9 km MLD data centered on coordinates from GPS taken while sampling. Mixed layer depths from profiling floats and CTD measurements were obtained by the density method, using 0.03 kg m$^{-3}$ as the cutoff value from a reference depth of 10 m[25].

**Inter- and intra-station comparison.** To determine if each biomarker changed from the initial deeply mixed state at Station 4 during the Climax phase, samples from depths corresponding to 40 and 20% surface irradiance (below 25 m) were compared against the first day of occupation with Kruskal$-$Wallis test. To determine if each biomarker from in situ populations was significantly different between stations with different histories within the Climax and Decline phases, the Kruskal$-$Wallis test was also utilized.

**Statistics.** Kruskal$-$Wallis test ($\alpha = 0.05$) was used to determine statistical differences between values of accumulation rates, ROS staining, percent compromised membranes, relative lipid abundance, metacaspase/caspase enzyme activities, virus, TEP, and DOC and DOC$_{SA}$ concentrations across different phases of the phytoplankton annual cycle. Kruskal$-$Wallis was chosen throughout this analysis due to a lack of sample size homogeneity across bloom phases and stations. When

significant differences were found, a Dunn test with Holm $p$ value adjustment was performed as a multiple comparisons test between the bloom phases.

The general relationship between stratification and phytoplankton biovolume, cell, TEP, virus, and DOC and DOC$_{SA}$ concentration was shown using a LOESS analysis of best fit. The degree of vertical stratification and enrichment of phytoplankton, TEP, virus and DOC and DOC$_{SA}$ concentrations were assessed by fitting a general additive model (GAM) to each variable. Model optimization was done by fitting each variable to depth and determining the optimum number of knots and smoothing spline type (mgcv package version 1.8-31 in R) using the Akaike Information Criterion (AIC) as a guide for best fit. The exported $R^2$ and $p$ values were extracted from the optimum model. See Supplementary Table 2 for details.

**Reporting summary.** Further information on research design is available in the Nature Research Reporting Summary linked to this article.

## Data availability

Raw datasets of measured environmental parameters during the NAAMES cruises are accessible in the NASA SeaBass archive (https://seabass.gsfc.nasa.gov). Float profiler data are available from the University of Maine Float Explorer website for all NAAMES cruises (http://misclab.umeoce.maine.edu/floats/, accessed 9/9/2020). The physiological, environmental, and extracellular data generated in this study have been deposited in the Zenodo database under accession code 10.5281/zenodo.5512903. Source data are provided with this paper.

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

## Acknowledgements

This work was made possible by NASA's Earth Science Program in support of the North Atlantic Aerosol and Marine Ecosystem Study (15-RRNES15-0011 and 0NSSC18K1563 to K.D.B.; NNX15AF30G to M.J.B.), as well as with support from the National Science Foundation (OIA-2021032 to K.D.B., OCE-157943 to C.A.C., and OCE-1756254 to B.A.S.V.M.), the Gordon and Betty Moore Foundation (Award# 3789 to K.G.V.B.), and NASA's Future Investigators in Space Science and Technology program (FINESST; grant #826380 to K.D.B.; graduate support to BD). We thank the captain and crew of the R/V Atlantis for their tireless efforts and roles in helping to collect field samples and data during NAAMES. We also thank Kim Thamatrakoln, Chana Kranzler, and Austin Grubb for insightful conversations that helped guide this study, as well as Jo Hunter for technical assistance with lipid analyses.

## Author contributions

B.D. collected samples at sea for the Accumulation phase; processed and measured caspase/metacaspase, TEP, viruses, bacteria for all bloom phases; analyzed all data and led the writing of the manuscript. B.K. collected samples at sea for the Decline and Accumulation phases and helped write and edit the manuscript. C.T.J. collected samples at sea for the Decline phase, processed viruses and bacteria for the Decline phase, and edited the manuscript. C.P.L. collected samples at sea in the Winter Transition and Climax phases and edited the manuscript. K.G.V.B. helped with all statistical analyses and edited the manuscript. L.H. processed TEP samples for Winter Transition, Climax and Decline phases; helped process caspase/metacaspase samples for all phases; and edited the manuscript. F.N. processed virus and bacterial samples from Winter Transi- tion and Climax phases and helped to develop flow cytometry protocols used throughout the study. E.L.H. collected samples at sea in the Winter Transition and Climax phases, helped with analyzing TEP data, and edited the manuscript. S.J.K. collected samples at sea during the Accumulation phase, and analyzed pigment-based community abundance data throughout all NAAMES field campaigns; helped incorporate community abundance data; and edited the manuscript. L.B. collected, processed, and analyzed 16S rRNA-based community analyses for all NAAMES field campaigns; helped incorporate community analysis data; and edited the manuscript. D.P.L. and H.F. processed and analyzed lipid data and edited the manuscript. J.G. collected and analyzed in situ flow cytometry data throughout the NAAMES campaign and edited the manuscript. T.W. obtained and analyzed remote observation data of mixed layer depth and edited the manuscript. K.D.A.M. analyzed in situ mixed layer depth and buoyancy frequency data throughout the NAAMES campaigns and edited the manuscript. N.H. helped deploy, collect and analyze data from gliders throughout the NAAMES campaigns and edited the manuscript. N.B. collected, processed, and analyzed DOC samples at sea and for all NAAMES campaigns; helped analyze mixed layer depth data; and edited the manuscript. P.G. collected and analyzed mixed layer depth and buoyancy data throughout the NAAMES campaign and edited the manuscript. E.B., C.A.C., M.J.B., B.A.S.V.M., and K.D.B. provided NAAMES leadership and edited the manuscript. K.D.B. lead the overall study and also collected samples at sea for the Accumulation phase.

## Competing interests

The authors declare no competing interests.
