## [Peer Review File · Nature Communications]

Reviewer comments, first round review:

Reviewer #1 (Remarks to the Author):

The paper by Diaz et al. reports the dynamics and physiological states of the microbial community in the North Atlantic bloom (a relatively well-studied phenomenon/region, going back several decades, but still with many basic unknowns). The authors examine the bloom, primarily through the lens of the disturbance recovery hypothesis, that posits that accumulation (growth exceeds removal) starts early in winter upon deep mixing because of dilution of top-down pressures.

The manuscript shows indeed that there is a strong physiological difference between accumulation/climax phases and the decline, which is interesting. I supposed before reading the manuscript that perhaps a better understanding of the degree to which the decline is nutrient related vs top-down control related would be exposed (especially since there is a focus on viruses), but I don't think that's the case. For two reasons: 1.) because it's not clear to me that many of the metrics used (reactive oxygen species, lipid biomarkers, program cell death enzymes) are strongly indicative of particular processes – in particular e.g., virus infection, stress or death from nutrient limitation, photo-damage. 2.) It is also unclear to me that these biomarkers and cell death enzymes are primarily derived from phytoplankton, and not, heterotrophic protists, or even the predominant biomass in the system, bacteria. That doesn't diminish the observation that they are higher in the decline, but to me complicates the paper's focus from the phytoplankton perspective.

Notably, it appears "phytoplankton" refers throughout to eukaryotic phytoplankton, and not the numerically dominant phytoplankton (at least for some of the year, I assume), cyanobacteria. This should be noted, and then, I think, considered in regards to the discussion/conclusions. This again complicates that interpretation, because cyanobacteria are a 'significant portion of the biomass in the decline phase' (Line 171, based on Bolaños et al. 2020).

Relatedly, in general, a bit of information or context in regards to the phytoplankton community composition of these samples would be valuable. It might also help to determine if the very fast mixing event (Line 249). It seems likely these were different water masses moving past the study site, and, if so, were likely dominated by different phytoplankton communities, correct?

Figure 1, It's not clear to me what the lower case letters ("ab", "a") refer to in regards to statistical significance, perhaps these are conventions that I am not aware of. Also in other figures.

Figure 1, I don't understand the implications of the mixed layer depth going from 500m to >1000m very rapidly between sites. Is it that there is effectively no mixed layer detected in these very deep mixed layer depth profiles (> 1000m)? Perhaps inclusion of some of these profiles in the supplement might help clarify.

Figure 4 and DOC measurements (dissolved organic carbon). DOC was measured in GF/F filtered seawater (~0.7 µm glass filter), while it probably collects things smaller than that, but not systematically, does this mean that many bacteria and most viruses are included in the DOC measurement? How does that complicate correlating viral abundances to DOC concentrations?

Figure 6 and 7, A bit of discussion is warranted in regards to what degree is it likely that changes in community composition (perhaps from different water masses) contribute to these dynamics?

Supplemental Figure 2: Would be nice to also see the bacterial abundances. Is there a typo in the axis, second virus plot from the left? It appears this figure isn't cited in the text?

Reviewer #2 (Remarks to the Author):

This paper presents an analysis of data collected from several cruises as part of the NAAMES campaign and attempts to link the mixed layer dynamics with phytoplankton physiology. Unfortunately the analysis of the physical water column dynamics are lacking, many statements are vague, and the conclusions are not well supported. I do not think this functions well as a short

paper and I recommend that it be rejected from Nature Communications. I do think this will eventually be publishable in a journal where the authors will have more space to describe their methods and justify their conclusions.

The central conclusion of the paper is that "the physical water column dynamics are intimately tied to cell stress, programmed cell death, and viral predation." Despite this statement, the paper does not discuss the physical dynamics. Instead, timeseries of the mixed layer depth and buoyancy frequency are given (both of which are difficult to interpret as described below). One way to investigate the physical "dynamics" would be to determine whether changes in the stratification result from lateral advection or vertical mixing. Although it isn't trivial to diagnose lateral advection from CTD stations, it should be possible to disentangle these effects vertically-integrated heat and salt budgets and estimates of the surface heat and freshwater fluxes. Satellite imagery and current speed might also be useful.

Although it is nice that the observations span a large range of latitudes, a significant problem with the analysis of the dataset is that bloom phases are not uniformly sampled in space. For example, all of the points in the "accumulation" phase are south of about 44N, while all but one of the points in the "climax" phase are north of 48N. Since the phytoplankton communities, water column structure, and seasonal forcing are very different in these regions, as the data is presented it is impossible for the reader to disentangle the seasonal timing from locationally-dependent effects. It should be possible to analyze this further, but it would significantly lengthen the manuscript.

The definition for the mixed layer depth (which is not given in the paper but is in the methods) is non-standard and complicates the interpretation of the results. The methods section states that the mixed layer depth is defined based on the standard deviation of the potential density profile. Although it isn't clear exactly how the standard deviation is calculated, this definition will be sensitive to the stratification below the mixed layer base. This could prove problematic, particular given that the definition is used to compare profiles from regions with different vertical structures and at various times during the year.

Stylistically, this paper does not function well as a short paper. Many terms and metrics were undefined in the text to the extent where I found the text unreadable without reading the Methods section in parallel. I appreciate that the analyses that are presented are not simple, but I think the paper would be more effective in a longer format journal where all terms could be explicitly defined.

Minor comments

The mixed layer depth should be clearly defined in the main text.

The definition of the critical depth on line 61 is not clear and could lead to confusion with the compensation depth.

line 65: are you referring to a specific study which treats the loss rate as constant throughout the year?

line 90: "water column mixing dynamics" is vague and "dynamics" is probably redundant here.

line 108: why refer to 4-7 distinct water columns and not give the number?

The text and caption to figure 1 refer to "water masses". It isn't clear what this refers to, but it doesn't seem to refer to the standard definition which is a body of water with similar physical properties.

I don't believe the curves in Figure 1c and d are explained.

line 119: The sentence here is vague and hard to understand. What is statistically indistinguishable?

line 124: "water column buoyancy frequency" needs to be defined more precisely in the main text. I suspect that the measure that is used (an average of the buoyancy frequency in the upper 100m) is largely controlled by the mixed layer depth when the MLD is shallower than 100m and it isn't clear whether this metric adds anything beyond mixed layer depth. If it does (e.g. if stratification within the mixed layer is important) then this should be explored further.

Reviewer #3 (Remarks to the Author):

Review of NCOMMS-20-46890

Seasonal mixed layer dynamics shape phytoplankton physiology, viral infection, and accumulation in the North Atlantic

Overview:

In this manuscript, Diaz and colleagues attempt to generalize how mixed layer dynamics in the Western North Atlantic impact phytoplankton physiology, using field data collected as part of the North Atlantic Aerosol and Marine Ecosystems Study (NAAMES). The authors attempt to link general markers of phytoplankton cell stress, including intracellular reactive oxygen species (ROS), cell membrane integrity, programmed cell death (PCD) protease activity, and lipid composition (oxidized phosphatidylcholine, triacylglycerols, and betaine-like lipids) to phases of the North Atlantic bloom cycle, thereby inferring physical mixing impacts upon cell physiology. In addition, the authors investigated changes in water column dissolved organic carbon (DOC), transparent exopolymer particles (TEP), and viruses. The authors investigate these parameters in the context of the disturbance-recovery hypothesis with the goal of using physiological markers and virus concentrations to better understand phytoplankton accumulation and loss. While I appreciate the authors' interest in linking large-scale physical mixing features with general markers of phytoplankton physiology, one major weakness of the study is the failure to acknowledge potential impacts of changing community composition upon the measurements and interpretations.

Bolanas et al. (2020) described significant community composition changes observed between different stations and seasons in the NAAMES study. Of particular concern is the shift from cyanobacterial dominated communities to those with >25% diatoms and prymnesiophytes. Using measured virus abundance as an example of how community composition could impact data interpretation, virus burst sizes depend upon the host. Viruses infecting larger eukaryotic phytoplankton tend to have burst sizes an order of magnitude larger than those infecting cyanobacteria. Could the changes in virus cell-1 be explained by community composition changes? Similarly, how much would you expect community composition to impact TEP production? With regards to physiological states, how universal are the markers used (oxidized phosphatidylcholine, triacylglycerols, betaine-like lipids, caspase and metacaspase activity) to the diverse assemblage of phytoplankton observed over the spatial and temporal scale of the study?

Specific comments:

Line 76-82 - All references cited are for eukaryotic phytoplankton, either include cyanobacteria, or be more specific about type of phytoplankton these processes apply to and relevance to ecosystem studied.

Lines 259-261 - "The sudden transition to lower irradiance may have triggered some phytoplankton to undergo stress-induced viral infection and TEP production". The only citation is

for a recent study on temperate viral infections in *Emiliana huxleyi*, but based on Bolanos et al. (2020), it is unlikely these were the dominant member of the phytoplankton community. Do you have other evidence that lysogenic viruses enter lytic phase upon light limitation? Also, would you expect less TEP production with lessening of irradiance?

Lines 266-269 - TEP and virus concentrations in the surrounding waters increased consistent with nutrient stress and/ or viral infection. Wouldn't re-stratification also increase irradiance exposure, a potential cause of TEP production?

Lines 271-274 - Seemed highly speculative.

Lines 289 - 310 - "compared two water masses in the decline phase" - How did the communities compare between the subpolar (Decline Station 6) and subtropical (Decline Station 2)?

Line 320 - "prolonged stratification" - what is the time scale for "prolonged"?

Lines 325-326 - "These decline phase biomarkers have been extensively linked to cell death and late stages of viral infection in diverse model systems in culture". Metacaspase orthologs have not been found in marine *Synechococcus* or *Prochlorococcus* (Bidle, 2015 and references within). The manuscript considered here should include discussion on how the relative abundance of these non-diazotrophic cyanobacteria might impact biomarker comparison among stations and the resulting interpretations of community physiology.

Methods:

Lines 379-397 - Viruses were quantified by flow cytometry, but it is often difficult to distinguish bacteriophage from instrument noise. I would like to see an illustrative flow cytometry plot in the supplementary information.

I have concerns about viruses and TEP accumulation calculated from deckboard incubations while plankton accumulation rates were calculated from 5m depth in situ sampling. The authors note that the 5m depth phytoplankton accumulation rates were not statistically different from the in situ accumulation rates (Fig. 1c, 1d), but it seems simpler to use the same treatments for accumulation rates of phytoplankton, viruses, and TEP. Is there a compelling reason to use the 5m depth values?

Figures:

A map of transects and water masses would be an excellent Supp. Fig to orient the reader (e.g. Fox et al. 2020 Fig. 1)

Citations:

Bidle, K.D. The molecular ecophysiology of programmed cell death in marine phytoplankton. *Ann. Rev. Mar. Sci.* 7, 341-375.

Bolaños, L. M. et al. Small phytoplankton dominate western North Atlantic biomass. *ISME J.* 14, 1663–1674 (2020).^[1]

Fox, J. et al. Phytoplankton growth and productivity in the western North Atlantic: Observations of regional variability from the NAAMES field campaigns. *Front. Mar. Sci.* 7, (2020)
DOI=10.3389/fmars.2020.00024.

Reviewer #4 (Remarks to the Author):

In the current manuscript by Diaz and colleagues, authors aim to link the effects of surface water stratification to phytoplankton dynamics and physiology within the same layer. By investigating water mixing by "season" along with phytoplankton abundance, oxidative stress levels (proxied by ROS concentration, membrane integrity, OxPC, and TEP), TAG and BLL levels, and caspase and metacaspase activity, authors were able to suggest an intimate connection between cell stress, death, and viral predation. Authors go further to suggest the current findings support the disturbance-recovery hypothesis.

As I was invited specifically for expertise in lipidomics, I'll focus the remaining comments on this portion of the text:

The Bligh and Dyer method of lipid extraction along with normal phase LCMS-MS are standard techniques for identifying lipids. While RT and fragmentation pattern matching are acceptable routes towards identification of a reasonable level, with so few lipids discussed in the manuscript I was surprised that no supplemental figures or tables were included relating to their identification. Authors report that they will include this raw data via GitHub after publication but it would be sufficient to say that these data could be reasonably displayed in a single figure (in the supplement) to convince the reader that the appropriate identifications were made.

Additionally, in figure 3a, there appear to be two replicates for the winter transition that could easily be statistical outliers. Was this addressed statistically and what would be the interpretation for why two replicates had such drastically high concentrations ($\sim 4\times$) O_xPC relative to other samples of the same season and is there a possible explanation as to why there were no other replicates like this for the other seasons? Other related questions could be asked of the potential outliers in 3b, 3c, and 3d.

A final unrelated comment: there is very little discussion related to the breakdown of phytoplankton species/groups present during sampling (i.e. ratio of dinoflagellates to diatoms to bacteria, etc., does a single species dominate at a specific sample point or did a bloom occur during a specific sampling event). It could be interesting in terms of interpretation to discuss any of these events in relation to the trends observed in figure 5.

Tracking #: NCOMMS-20-46890

Seasonal mixed layer depth shapes phytoplankton physiology, viral production, and accumulation in the North Atlantic" by Diaz et al.

Response to Reviewers (*in blue italics*)

Reviewer #1 (Remarks to the Author):

The paper by Diaz et al. reports the dynamics and physiological states of the microbial community in the North Atlantic bloom (a relatively well-studied phenomenon/region, going back several decades, but still with many basic unknowns). The authors examine the bloom, primarily through the lens of the disturbance recovery hypothesis, that posits that accumulation (growth exceeds removal) starts early in winter upon deep mixing because of dilution of top-down pressures.

The manuscript shows indeed that there is a strong physiological difference between accumulation/climax phases and the decline, which is interesting. I supposed before reading the manuscript that perhaps a better understanding of the degree to which the decline is nutrient related vs top-down control related would be exposed (especially since there is a focus on viruses), but I don't think that's the case. For two reasons: 1.) because it's not clear to me that many of the metrics used (reactive oxygen species, lipid biomarkers, program cell death enzymes) are strongly indicative of particular processes – in particular e.g., virus infection, stress or death from nutrient limitation, photo-damage. 2.) It is also unclear to me that these biomarkers and cell death enzymes are primarily derived from phytoplankton, and not, heterotrophic protists, or even the predominant biomass in the system, bacteria. That doesn't diminish the observation that they are higher in the decline, but to me complicates the paper's focus from the phytoplankton perspective.

We appreciate Reviewer#1's general recognition of our work and finding interest in the strong physiological difference between accumulation, climax, and decline phases. We agree that these aspects of inherent and discernable differences in the physiology of phytoplankton cells over different bloom phases and in different water masses in relation to mixed layer states is unique and novel. Our work aims to address fundamental unknowns about microbial community dynamics in the North Atlantic bloom, which has indeed been of interest to the field for decades (as noted by Reviewer#1). At the same time, Reviewer#1 raises some important points regarding the interpretation of our data as it relates to the main take-home messages. We address these concerns below.

I supposed before reading the manuscript that perhaps a better understanding of the degree to which the decline is nutrient related vs top-down control related would be exposed (especially since there is a focus on viruses), but I don't think that's the case. For two reasons: 1.) because it's not clear to me that many of the metrics used (reactive oxygen species, lipid biomarkers, program cell death enzymes) are strongly indicative of particular processes – in particular e.g., virus infection, stress or death from nutrient limitation, photo-damage

We acknowledge (and agree) that macronutrient nutrient states associated with our sampled water masses during the different bloom phases is important to determine if the community state(s) was(were) influenced by 'bottom-up' processes, such as nutrient limitation. We interpreted the reviewer's comment "that perhaps a better understanding of the degree to which the decline is nutrient related vs top-down would be exposed" to also be more broadly questioning mechanisms of phytoplankton loss throughout the annual bloom and not specifically during the Decline phase. We have phrased our responses below accordingly.

We have now included macronutrient (nitrate/nitrite and phosphate) concentrations for water masses sampled across different water types (Gulf Stream and Sargasso Sea, sub-tropical, temperate, and sub-polar; as defined by Penna and Gaube 2019) during the different bloom phases as they relate to mixed layer stratification (**Supplementary Fig. 3**). Our data first shows that the overall nutrient levels depended on water type with Subpolar stations had generally higher concentrations. Furthermore, it was clear that nutrient levels during the Climax phase, in which populations were characterized by high levels of ROS, TEP and oxidized PCs, were similar (or higher; $\text{NO}_3/\text{NO}_2 > 5 \mu\text{M}$; $\text{PO}_4 > 0.2 \mu\text{M}$) than those observed for Accumulation phase samples, which had notably lower oxidative stress signatures. This argues against nutrient limitation causing the observed oxidative stress responses during Climax phase. (**Lines 185-190**)

We also performed PCA across data from individual stations throughout the bloom (**Supplementary Figure 9**), and within the Climax phase (**Supplementary Figure 10**). This confirmed that the observed ROS, TEP, and oxidized PC signatures were likely not driven by nutrient limitation, as they did not negatively covary with macronutrient concentration; if the vectors had pointed in opposite directions, it would have implicated nutrient stress. Although lower ROS was found during decline phase, in this bloom phase it actually positively varied with nutrient limitation, suggesting that nutrient draw down contributed to relatively elevated ROS during this phase (**Supplementary Figure 11**). We have now incorporated interpretive discussion of these observations in the main text (**Lines 295-319**).

Notable nutrient drawdown was observed during Decline phase (as expected), although sub-polar stations had detectable ($\sim 2 \mu\text{M}$) nitrate/nitrite concentration, suggesting that macronutrients were not limiting. Accordingly, phytoplankton were in high abundance at the sub-polar Decline phase station (Decline Station 6). We've added discussion of macronutrient variability within the Decline phase (**Lines 403-406**), and their potential connection to community composition (**Lines 406-414, 419-426**).

Reviewer 1 also raised the issue that some of our metrics (e.g., ROS, lipid biomarkers, PCD proteolytic enzymes) are not individually diagnostic of particular stress/death processes, such as virus infection, nutrient limitation, and photo-damage. Indeed, these markers represent core cell stress and death processes and are shared among various biotic- and abiotic-stress processes, this has been shown extensively in the literature (as referenced within our paper). They cannot discriminate between different forms of stress/death. They are explicitly tied to cell fate in response to various stressors, which is why we focus on them. We summarize this rationale in **Lines 173-180**, with the goal of framing how to interpret these biomarkers to the reader. We hope that our work provides a novel base of knowledge upon which other researchers can build and better elucidate the specific cellular mechanisms behind these stresses and to what extent these stresses are shared among the mixed community.

2.) It is also unclear to me that these biomarkers and cell death enzymes are primarily derived from phytoplankton, and not, heterotrophic protists, or even the predominant biomass in the system, bacteria. That doesn't diminish the observation that they are higher in the decline, but to me complicates the paper's focus from the phytoplankton perspective.

Reviewer#1 correctly points out that these biomarkers and cell death enzyme activities are not explicitly derived from phytoplankton. They are indeed also present in heterotrophic protists, and in some cases, can derive from bacteria (e.g., a majority of the viruses detected in our flow cytometry measurements are likely bacteriophage). We appreciate Reviewer#1 recognizing that it does not diminish that markers are notably higher in either Climax or Decline, as they are indicative of fundamental cellular changes experienced by resident microbes. We acknowledge that it does complicate the interpretation away from the sole focus on phytoplankton. The purpose of our paper is not to resolve to what extent specific taxa are stressed via nutrient limitation, virus infection, etc. but rather to characterize the physiological changes associated

with mixed phytoplankton populations in relation to mixed layer depth and stratification. We address this comment in several ways. First, we now clarify the organisms and size ranges from which the intracellular biomarkers likely derive (main text, **Lines 160-172**). Second, we've now included a new **Supplementary Table 1** which further clarifies what organisms are contributing to both cellular and environmental/ecosystem biomarkers. This will help the reader understand the likely processes that are connected to our measurements and provide context for our conclusions.

Reviewer#1 is correct that some biomarkers such as caspase activity would include heterotrophic protist grazers. To date, caspase activities have been found in diverse phytoplankton (coccolithophores, diatoms, chlorophytes, dinoflagellates, all of which were present in NAAMES communities) that are only undergoing stress/dying they are not found in healthy cells. Caspase activity induction is similarly specific to stressed or dying protists, for the few who have been investigated (e.g., Trypanosomes). They haven't been specifically investigated in diverse marine protists, but it is logical to infer that caspase activities are also induced under stress/dying cells in these systems too. We now add a broader interpretive context for our markers as they relate to the microbial system that is experiencing changes in mixed layer depth across the seasonal bloom. We argue that the fact that our measurements derive from a variety of microbial sources within the community actually strengthens and expands our central message regarding physiological state changes of a 'biological system' within the mixed layer throughout a bloom. A main point from our study is that the system itself is characterized by discernable physiological states related to mixed layer dynamics. Some measurements (ROS/compromised membranes) are indeed specific for eukaryotic phytoplankton. For example, measurements and analyses of ROS (H2-DCFDA) and compromised membrane (SYTOX staining) came from eukaryotic phytoplankton since we could use size and chlorophyll content in our flow cytometry methods to differentiate these cells. We now discuss this extensively in (**Lines 160-180**). Taken together, our findings reveal that most of the system of eukaryotic microbes was experiencing unique stresses in each bloom phase, especially the Climax and Decline phases.

We feel that we are justified to say that the system experienced each stress since we normalized each metric. For example, OxPC was normalized to total phosphatidylcholine extracted, which was derived from eukaryotic phytoplankton and grazers (**Supplementary Table 1**). While extensive laboratory studies have not been done on the cellular mechanisms related to changes in biomarkers for most heterotrophic protists, it is conceivable (if not likely) that their food source (phytoplankton) was reducing in numbers and they themselves were experiencing resource limitation and associated stress/death. This is consistent with observations of induction of these pathways. We now include some discussion that: "Additional size fractions, high speed cell-sorting, and/or single-cell analyses, combined with nucleic acid-based techniques would provide finer resolution on which taxa experienced specific physiological changes and paint a more comprehensive picture of virus infection dynamics throughout the North Atlantic bloom." (**Lines 340-343**)

To also help address one of Reviewer#1's general concerns, regarding shedding light on the influence of top-down vs. bottom-up contribution to the decline, we performed a new analysis in **Fig. 6**, which clusters nutrients, biomarkers, and accumulation by station, with a map of their geographical location/ sub-region in the North Atlantic. This new analysis revealed for the first time that groups of biomarkers cluster together by bloom phase. The emergence of these community stress patterns for phytoplankton communities across bloom phases and geographical locations is unprecedented. (**Fig. 6 and Lines 279-294**). It also revealed that the high levels virus cell⁻¹ was found throughout the Decline phase, regardless of latitude, suggesting that high virus cell⁻¹ is a persistent ecosystem pressure even though latitude/nutrient limitation might be variable. (**Lines 424-426**).

We have also added macronutrients to **Fig. 7**, which cluster nutrients, biomarkers, and phytoplankton community type by bloom phase and station, respectively. This allowed us to further characterize how high levels of ROS, TEP and oxidized PCs, which have been linked to cell death and stress resulting in lower accumulation/decline, were probably not related to the bottom-up process of nutrient limitation during the Climax phase. PCA across data from individual stations throughout the bloom (**Fig. 7**), and within the Climax phase (**Supplementary Fig. 10**) confirmed that the observed ROS, TEP, and oxidized PC signatures in the Climax phase were not driven by nutrient limitation, as they did not negatively covary with macronutrient concentration; if the vectors had pointed in opposite directions, it would have implicated nutrient stress. Although lower ROS was found during Decline phase, in this bloom phase it negatively covaried with nutrient concentration, suggesting that nutrient limitation, when present, contributed to elevated ROS within this phase (**Supplementary Fig. 11**). We have now incorporated interpretive discussion of these observations in the main text. (**Lines 307-316**)

Notably, it appears “phytoplankton” refers throughout to eukaryotic phytoplankton, and not the numerically dominant phytoplankton (at least for some of the year, I assume), cyanobacteria. This should be noted, and then, I think, considered in regards to the discussion/conclusions. This again complicates that interpretation, because cyanobacteria are a 'significant portion of the biomass in the decline phase' (Line 171, based on Bolaños et al. 2020).

We agree with the reviewer’s point here. Many of our measurements, and much of our interpretive perspective in the paper, is geared towards eukaryotic phytoplankton. Reviewer#1 is correct that this influences our interpretations, discussion, and conclusions and should be adequately noted. We have clarified our discussion of phytoplankton dynamics to specify eukaryotic phytoplankton and, where appropriate, have included context for possible cyanobacteria presence in the community and whether our markers would have included them. Reviewer#3 and Reviewer#4 also commented on community composition. We incorporated community composition analyses as **Fig. 7** and **Supplementary Fig. 10** and **11**, using data from two independent methods (16S rRNA sequencing from Bolanos 2021⁸ and HPLC of pigments from Kramer et al 2020⁹) to better characterize how community structure related to our biomarkers. We analyzed to what degree our biomarkers covaried with community composition across all bloom phases (**Fig. 7**) and also across stations within the Climax and Decline phases (**Supplementary Fig. 10, 11**). Our bloom-wide analyses found that cyanobacteria taxa positively covaried with higher levels of stratification, virus cell⁻¹, TAG ChIA⁻¹, caspase activity, compromised membranes and DOC_{SA} (**Fig. 7**). We now discuss this in the main text (**Lines 295-306**). We provide arguments as to why the TAG, caspase, or compromised membranes are likely not from cyanobacteria biomass, and why the viruses are likely not from cyanobacteria phages. We lay out our logic in **Lines 327-339**:

“Although cyanobacteria were associated with compromised membranes, TAGs, caspase activity or viruses in the highly stratified waters of the Decline phase, cyanobacteria were likely not the source of these biomarkers several reasons. First, cyanobacteria did not positively covary with compromised membranes and caspase activity; likewise, they only weakly covaried with viruses cell⁻¹ (**Supplementary Fig. 11c, 11d**). Second, our compromised membrane detection procedure did not include cyanobacteria due to differentiations in cell size (**Supplementary Table 1**). Third, TAGs are not known to be a major component of marine cyanobacteria biomass³³. They are more likely the product of eukaryotic phytoplankton⁴²⁻⁴⁴ or zooplankton metabolism⁶⁷ in a nutrient limited environment. Fourth, caspase/metacaspase orthologs have not been identified in the genomes of unicellular cyanobacteria, such as *Prochlorococcus* and *Synechococcus*⁶⁸. Lastly, virus production and cell death would mechanistically reduce host cell concentrations so that higher cyanophage abundance would be associated with lower cyanobacteria host populations.”

In performing our edits and response to reviewers, we did note that Bolaños et al. 2020⁷ did not actually report on the decline phase (NAAMES #3). Rather, Bolaños et al. 2021⁸ reported on community analysis for all bloom phases and NAAMES cruises so we have now included both references (and datasets) in our analyses and discussion.

Relatedly, in general, a bit of information or context in regards to the phytoplankton community composition of these samples would be valuable. It might also help to determine if the very fast mixing event (Line 249). It seems likely these were different water masses moving past the study site, and, if so, were likely dominated by different phytoplankton communities, correct?

*We agree that information about community composition is important to consider, given our analyses span different water masses across different seasons and represent distinct phytoplankton communities. We appreciate Reviewer#1 for raising this issue. We performed comparative analyses between our markers and two independent, published measures of community composition—16S rRNA sequencing (using the plastid sequences to discern eukaryotic taxa; Bolaños et al. 2021⁸) and HPLC pigment analysis (Kramer et al. 2020⁹) — to infer taxa identities. We analyzed to what degree our biomarkers covaried with community composition across all of the bloom phases (**Fig. 7**) and also across populations within either the climax and decline phases (**Supplementary Fig. 10, 11**). Our bloom-wide, analyses found that cyanobacteria taxa positively covaried with higher levels of stratification, virus cell⁻¹, TAG ChlA⁻¹, while the diatom taxa positively covaried with phosphate and nitrate/nitrite levels, TEP cell⁻¹, and oxidative stress markers (**Fig. 7**). We now discuss how community composition was related to our biomarkers throughout the bloom in **Lines 295-306**, and within the decline and climax phases in **Lines 320-327**.*

*We have also added discussion on community composition of the case studies (**Fig. 9**) and how the community types at the different stages in the bloom could be related to degree of stratification within the Climax phase, (**Lines 388-395**), and nutrient status and geographic location within the Decline phase (**Lines 406-414**). We also added a discussion paragraph about the correlations between diatoms and oxidative stress/ TEP cell⁻¹, and cyanobacteria and Decline phase biomarkers. The paragraph concludes with suggestions of how to improve sampling to get a clearer picture of “who” was experiencing stress/dying, and “who” was growing in different stages of the bloom (**Lines 320-343**).*

*Based on data published in Bolanos et al 2020⁸ for Climax Station 4, the community compositions between the recently mixed and subsequently stratified conditions were largely the same (see: Fig. 3, late spring, Station 4) which argues against different water masses containing distinct communities moving into/past the study site (**Lines 351-352**).*

Figure 1, It's not clear to me what the lower case letters (“ab”, “a”) refer to in regards to statistical significance, perhaps these are conventions that I am not aware of. Also in other figures.

We updated all figure legends using this convention to include descriptions of what multiple letters denote in our statistical analysis. They now contain the statement: “Different letters denote statistically significant groups ($p < 0.05$, Kruskal-Wallis test with Dunn corrections for multiple comparisons). Intergroup comparisons with more than one letter denote no significant difference between the two groups.”

Figure 1, I don't understand the implications of the mixed layer depth going from 500m to >1000m very rapidly between sites. Is it that there is effectively no mixed layer detected in these very deep mixed layer depth profiles (> 1000m)? Perhaps inclusion of some of these profiles in the supplement might help clarify.

*We apologize for the confusion in this part of **Fig. 1b**. Our intention was to show a composite plot of all the recorded mixed layer depths from float measurements that were*

deployed in all water masses sampled. In going back through our data, we noted that the profiling float that recorded/reported the very deep mixed layer depths >1000 m was in subpolar, Labrador Sea water. Consequently, this site was inconsistent with the water types and mixed layer depths from all other sites sampled in the Northeast Atlantic study area. We have removed the float data from this outlier water mass and have revised **Fig. 1b** and accordingly. We have also updated Figure 1A to better show the locations of stations sampled in the during the NAAMES expeditions in relation to different sub-regions (Gulf Stream and Sargasso Sea, Subtropical, Temperate, and Subpolar; as defined by Penna & Gaube, 2019). We have also added an additional Supplementary figure (**Supplementary Fig. 1**) that shows the tracks of profiling floats used for our analyses along within the NAAMES sample area.

Figure 4 and DOC measurements (dissolved organic carbon). DOC was measured in GF/F filtered seawater (~0.7 µm glass filter), while it probably collects things smaller than that, but not systematically, does this mean that many bacteria and most viruses are included in the DOC measurement? How does that complicate correlating viral abundances to DOC concentrations?

*Reviewer#1 is correct that our DOC measurements were performed on GF/F-filtered seawater, which is a standard way to operationally define DOC in the oceanographic community. It would indeed allow for essentially all viruses to pass through into the filtrate. Many bacteria (~60%) also pass through this pore size but, given they encompass a range of sizes and can be associated with particles, it is unlikely that all bacteria would be represented in filtrates. We generated plots comparing total DOC and accumulated DOC (DOC_{SA}) pools with and without bacteria and viruses (**Supplementary Fig. 6**) in order to determine the contribution of viruses and bacteria to the background DOC and DOC_{SA} signatures. Our calculations used our measured bacteria and virus concentrations and carbon quotas of 12.4 fg bacterial cell⁻¹ (Fukuda et al 1998¹⁰) and 0.2 fg virus particle⁻¹ (Jover et al 2014¹¹), which is on the conservative side of a range of reported values (another reported value was 0.055 fg virus particle⁻¹; also reference in Jover et al 2014¹¹). Our analysis showed that the combined bacteria and virus pools had negligible contributions to both DOC and DOC_{SA}, as the microbe-corrected 95% confidence interval falls on the 1:1 line; there was no significant difference detected between the two (**Supplementary Fig. 6**). Hence, the prominent increase in DOC_{SA} observed during the 'Decline' phase could not be attributed to increases in bacteria or virus concentrations. We discuss this in the main text in **Lines 230-243**, and in the methods in **Lines 619-626**.*

Figure 6 and 7, A bit of discussion is warranted in regards to what degree is it likely that changes in community composition (perhaps from different water masses) contribute to these dynamics?

As noted above, we performed comparative analyses between our markers and two independent, published measures of community composition—16S rRNA sequencing (using the plastid sequences to discern eukaryotic taxa; Bolaños et al. 2021) and HPLC pigment analysis (Kramer et al. 2020) — to infer taxa identities.

*To address community composition at Climax Station 4, (previously **Fig.6**, now **Fig. 8**) analysis, we've added discussion with a reference to Bolanos et al 2021 which shows that the community abundance did not significantly change upon the transition from deep mixing to shallow mixed layer at Climax Station 4. We've also added discussion about community composition and some of the implications for our inter-station analysis, (previously **Fig. 7**, now **Fig. 9**). It reads:*

*“Based on pigment composition, both days of Climax Station 4 were classified as diatom community types. In contrast, Climax Station 1 had a higher relative abundance of diatoms on the first day of occupation and dinoflagellates were relatively more abundant on the second day of occupation. Based on 16S analysis, the most abundant group on Climax Station 4 for both days of occupation was *Ostreococcus* (37% & 44% relative abundance, R.A.), while Climax*

Station 1 contained mainly *Micromonas* (47% R.A.). This may indicate a succession to dinoflagellates or chlorophytes as the water column becomes more stratified during this bloom phase.” (Lines 388-395)

We have included community composition in our principle component analysis (Fig. 7) but not in the heatmaps of Figs. 8 and 9 (previously Figs. 6 and 7) since we did not have the same level of replication as our multifaceted biomarker data, and would be unable to make similar statistical comparisons. At each station, our group focused on 1-2 sampling points per day, with replicates within these sampling points, which is what we present in most of this paper. Collaborating NAAMES groups sampled for community analysis with less replication at each collection, but with more discrete times sampled throughout the day. Hence, the community analysis data and our biomarker data overlapped for one or two data points for each sampling day. Since Fig. 8, 9 represent statistical comparisons, we felt it was inappropriate to include samples with fewer replicates in the heatmap.

Likewise, the new heatmap presented in Fig. 6 represents the median values of many samples, so plotting values from 1-2 replicates of community composition in the same figure would not be as robust as the biomarkers shown, which have more replicates to pool the median value from. We felt that combining the community composition data with the biomarker data was important, so we have now done so in our PCA analysis in Fig. 7, Supplementary Fig. 10, 11, and in the main text where appropriate.

Supplementary Figure 2: Would be nice to also see the bacterial abundances. Is there a typo in the axis, second virus plot from the left? It appears this figure isn't cited in the text?

We agree and have now included plots of the vertical profiles of bacteria concentration and corresponding enrichment analysis via GAM in what are now Supplementary Fig.'s 8 and 9. We have corrected the typo in the axis of that figure and the non-text citation.

References cited:

1. Jónasdóttir, S. Fatty Acid Profiles and Production in Marine Phytoplankton. *Mar. Drugs* **17**, 151 (2019).
2. Lombardi, A. T. & Wangersky, P. J. Particulate lipid class composition of three marine phytoplankters *Chaetoceros gracilis*, *Isochrysis galbana* (Tahiti) and *Dunaliella tertiolecta* grown in batch culture. *Hydrobiologia* **306**, 1–6 (1995).
3. Brembu, T., Mühloth, A., Alipanah, L. & Bones, A. M. The effects of phosphorus limitation on carbon metabolism in diatoms. *Philosophical Transactions of the Royal Society B: Biological Sciences* vol. 372 (2017).
4. Hunter, J. E., Frada, M. J., Fredricks, H. F., Vardi, A. & Van Mooy, B. A. S. Targeted and untargeted lipidomics of *Emiliana huxleyi* viral infection and life cycle phases highlights molecular biomarkers of infection, susceptibility, and ploidy. *Front. Mar. Sci.* **2**, 81 (2015).
5. Lee, R., Hagen, W. & Kattner, G. Lipid storage in marine zooplankton. *Mar. Ecol. Prog. Ser.* **307**, 273–306 (2006).
6. Bidle, K. D. The Molecular Ecophysiology of Programmed Cell Death in Marine Phytoplankton. *Ann. Rev. Mar. Sci.* **7**, 341–375 (2015).
7. Bolaños, L. M. *et al.* Small phytoplankton dominate western North Atlantic biomass. *ISME J.* **14**, 1663–1674 (2020).
8. Bolaños, L. M. *et al.* Seasonality of the Microbial Community Composition in the North Atlantic. *Front. Mar. Sci.* **8**, 23 (2021).
9. Kramer, S. J., Siegel, D. A. & Graff, J. R. Phytoplankton Community Composition Determined From Co-variability Among Phytoplankton Pigments From the NAAMES Field Campaign. *Front. Mar. Sci.* **7**, (2020).
10. Fukuda, R., Ogawa, H., Nagata, T. & Koike, I. Direct determination of carbon and

- nitrogen contents of natural bacterial assemblages in marine environments. *Appl. Environ. Microbiol.* **64**, 3352–3358 (1998).
11. Jover, L. F., Effler, T. C., Buchan, A., Wilhelm, S. W. & Weitz, J. S. The elemental composition of virus particles: Implications for marine biogeochemical cycles. *Nat. Rev. Microbiol.* **12**, 519–528 (2014).

Reviewer #2 (Remarks to the Author):

This paper presents an analysis of data collected from several cruises as part of the NAAMES campaign and attempts to link the mixed layer dynamics with phytoplankton physiology. Unfortunately the analysis of the physical water column dynamics are lacking, many statements are vague, and the conclusions are not well supported. I do not think this functions well as a short paper and I recommend that it be rejected from Nature Communications. I do think this will eventually be publishable in a journal where the authors will have more space to describe their methods and justify their conclusions.

We acknowledge Reviewer#2's concerns about our analyses of water column dynamics and perception that our discussions contained vague statements. We have taken considerable efforts to further clarify our analyses of mixed layer depths and strengthen our conclusions from the data presented. We have specifically revised our statements of water column 'dynamics', recognizing that our sampling (and associated analyses) of water masses captured microbial communities at discrete mixed layer depths and did not explicitly track the mixed layer dynamics within specific water masses. We have updated the main text so it states that we specifically examined mixed layer depth or stratification instead of the more vague "physical water column dynamics." We note that the Editor has kindly given us permission to expand the main text and now use this allowance to incorporate more detailed interpretive discussion in the main text and we hope that this helps address Reviewer#2's concerns.

The central conclusion of the paper is that "the physical water column dynamics are intimately tied to cell stress, programmed cell death, and viral predation." Despite this statement, the paper does not discuss the physical dynamics. Instead, timeseries of the mixed layer depth and buoyancy frequency are given (both of which are difficult to interpret as described below). One way to investigate the physical "dynamics" would be to determine whether changes in the stratification result from lateral advection or vertical mixing. Although it isn't trivial to diagnose lateral advection from CTD stations, it should be possible to disentangle these effects vertically-integrated heat and salt budgets and estimates of the surface heat and freshwater fluxes. Satellite imagery and current speed might also be useful.

We appreciate Reviewer#2's comments here. Upon reflecting and discussing this issue with our co-authors, we agree that we did not examine a wide breadth of the physical dynamics of water column processes, such as if stratification was due to lateral advection or vertical mixing. Instead, our analysis is relating two specific physical metrics of the water column, namely mixed layer depth and degree of stratification for stations sampled across seasonal bloom states as they relate to physiological biomarkers of resident phytoplankton. We revised our wording through the main text and in the conclusion to more specifically reflect that. For example: "Our findings provide novel evidence that seasonal stratification of the water column guides intracellular stress, programmed cell death, and viral predation." (Lines 428-429)

With regards to the deep mixing event at Climax Station 4 and rapid re-stratification that followed (now Fig. 8), we acknowledge that lateral advection of a different water mass may have possibly been the mechanism behind the stratification. Arguably, this would have been detectable as a significant change in community composition. However, we note that the phytoplankton community composition remained largely the same as shown by Bolanos et al

2021, so we feel justified to describe the changes in physiology are based on shallowing of the mixed layer depth, not via transport of a novel community into our sampling location via lateral advection. We note in the main text that the community composition stayed the same and provide a reference to that study (**Lines 351-352**).

Although it is nice that the observations span a large range of latitudes, a significant problem with the analysis of the dataset is that bloom phases are not uniformly sampled in space. For example, all of the points in the "accumulation" phase are south of about 44N, while all but one of the points in the "climax" phase are north of 48N. Since the phytoplankton communities, water column structure, and seasonal forcing are very different in these regions, as the data is presented it is impossible for the reader to disentangle the seasonal timing from locationally-dependent effects. It should be possible to analyze this further, but it would significantly lengthen the manuscript.

Reviewer#2 raises an important point about how geographic location (latitude) of sampled populations might impact their bloom phase. This was indeed an important consideration in the design of the NAAMES campaigns. A targeted sampling approach was used for phytoplankton populations based on satellite observations and the location of previously deployed optical profiling assets, which recorded water column properties. This provided key context in which to interpret phytoplankton responses to mixed layer changes. Hence, our sampling strategies had this fundamental, yet essential, constraint.

*We have now revised **Fig. 1a** to include a map overlaying our sampled water masses (stations) with broad water type classifications, as described by Penna and Gaube, (2019). This addition makes it clearer that all four field campaigns collected samples within the subtropical region and all but the accumulation phase (due to weather) collected samples in the Temperate and Subpolar regions. For the winter transition and climax phases, the timing of these events changed with latitude, so the latitudinal transect executed during NAAMES ensured a range of transition states. For the accumulation phase, this condition is simultaneously occurring across the NAAMES study site (albeit at different levels of development with latitude), so while the geographic extent of NAAMES data during this phase is more limited, our expectation is that our results are representative of this phase.*

*We also provide analyses of macronutrient concentrations broken out by bloom phase and water type (**Supplementary Fig. 3**). This includes a new analysis of the median values of our biomarkers, accumulation rates, and nutrients within the mixed layer across bloom phases and stations sampled across latitude and sub-regional water types (**Fig. 6**). Our analysis showed that the general biomarker patterns also held true across NAAMES stations when normalized to the highest and lowest median values throughout the bloom. Stations within a season tended to cluster together for notable marker signatures (virus cell⁻¹, OxPC, caspase activity, TAGs) across different latitudes suggesting that the seasonal imprint on water column stratification, bloom dynamics and community physiological state was more prominent than that of latitudinal location. This analysis shows that latitude does have an effect in terms of macronutrients and some biomarkers (ROS, metacaspase activity), but that each station within a bloom phase clusters more strongly to stations at different latitudes within that bloom phase than with stations from other bloom phases. Similarly, it showed that certain biomarkers clustered into distinct groups throughout the bloom. (**Lines 279-294**).*

*To further disentangle how station location/seasonal timing may have influenced our biomarkers and community type within a bloom phase, we now include PCA analysis between biomarkers, stratification, nutrients and community type across NAAMES (**Fig. 7**) and within Climax and Decline phases (**Supplementary Fig. 10 and 11**). Community analyses used data from two independent methods (16S rRNA sequencing from Bolanos et al 2021 and HPLC of pigments from Kramer et al 2020) to better characterize how community structure related to our biomarkers. We've included specific community types/ relative abundance per day in the main*

text accompanying our more detailed inter-station comparisons from Fig. 9. (Lines 296-306, Lines 320-343, Lines 388-395, Lines 406-414).

The definition for the mixed layer depth (which is not given in the paper but is in the methods) is non-standard and complicates the interpretation of the results. The methods section states that the mixed layer depth is defined based on the standard deviation of the potential density profile. Although it isn't clear exactly how the standard deviation is calculated, this definition will be sensitive to the stratification below the mixed layer base. This could prove problematic, particular given that the definition is used to compare profiles from regions with different vertical structures and at various times during the year.

To simplify comparison among MLDs and interpretation of the results, we've recalculated the MLD using the standard method, updated our method section appropriately, and noted in the main text how we calculated MLD. The MLD in this study was calculated using a density difference threshold of 0.03kg m^{-3} . (Lines 122-123, Lines 633-643)

Stylistically, this paper does not function well as a short paper. Many terms and metrics were undefined in the text to the extent where I found the text unreadable without reading the Methods section in parallel. I appreciate that the analyses that are presented are not simple, but I think the paper would be more effective in a longer format journal where all terms could be explicitly defined.

*We are sorry to hear that the reviewer found terms and metrics undefined and that this caused confusion and difficulty in reading our manuscript. We have updated the main text to introduce more clearly each of the biomarkers used (Lines 160-172), and now include a new **Supplementary Table 1** that outlines the size classes and organisms relevant to each biomarker, along with key publications that used the very similar or the same methods and reagents. We have expanded the discussion of the paper with new analyses and additional text to better define our terms and adequately discuss the results (as encouraged by the Editor) and hope that this alleviates Reviewer#2's concerns in this area.*

Minor comments

The mixed layer depth should be clearly defined in the main text.

We now clarified what the mixed layer is in the introduction, by separating the MLD definition into a separate sentence that reads; "--the uppermost region in the water column which is homogenized by convective and turbulent mixing." (Lines 62-63)

We also now include the method we used to calculate the MLD in the main text (Lines 122-123).

The definition of the critical depth on line 61 is not clear and could lead to confusion with the compensation depth.

We updated this sentence and the one prior to specialty mention how more light per day is available when the mixed layer depth is shallower, and that the critical depth refers to a "critical" MLD, instead of a "critical" depth, as we had phrased it before. This has been revised to read: "The critical depth hypothesis posits that, as the MLD shallows in the spring, phytoplankton photosynthetic rates increase in response to the increasing availability of daily irradiance. A "critical" MLD exists where photosynthetic rates can overcome respiratory losses, allowing for increased division rates and biomass accumulation⁵." (Lines 67-68)

line 65: are you referring to a specific study which treats the loss rate as constant throughout the year?

Yes, it was referring to the same reference as the previous sentence. We have included this reference again in this sentence to clarify and constant loss rates were used in the formulations of this study (**Line 71**).

line 90: "water column mixing dynamics" is vague and "dynamics" is probably redundant here.

We agree and have taken out the phrase (mixing dynamics) and replaced it with a more specific term such as "mixed layer depth" or "stratification" throughout the paper.

line 108: why refer to 4-7 distinct water columns and not give the number?

Each cruise sampled a different number of water columns (stations) so we provided the range of stations sampled. We note that the actual number of stations we sampled is presented in **Figure 1a**.

The text and caption to figure 1 refer to "water masses". It isn't clear what this refers to, but it doesn't seem to refer to the standard definition which is a body of water with similar physical properties.

We were referring to "water masses" as water associated with each station sampled. As noted, CTD and profiling float deployments characterized water column properties for these water masses. We revised the verbiage to describe the places we sampled as "locations, or 'stations'" (**Line 118**).

I don't believe the curves in Figure 1c and d are explained.

We included the descriptive phrase in the legend for **Fig. 1** (and similarly changed the legend for **Fig. 2**) stating: "Data in panels **c**) and **d**) are based on cell concentrations and contoured with ridgeline smoothing to represent the distribution of accumulation rates across stations within a given bloom phase. The size of contour peaks is driven by frequency of observations."

line 119: The sentence here is vague and hard to understand. What is statistically indistinguishable?

We have reworded this and now split it into two sentences. We are referring to the accumulation rates from each cruise, calculated from in situ and incubated populations. (**Lines 131-136**).

line 124: "water column buoyancy frequency" needs to be defined more precisely in the main text. I suspect that the measure that is used (an average of the buoyancy frequency in the upper 100m) is largely controlled by the mixed layer depth when the MLD is shallower than 100m and it isn't clear whether this metric adds anything beyond mixed layer depth. If it does (e.g. if stratification within the mixed layer is important) then this should be explored further.

We now define buoyancy frequency in the main text. It reads: "We quantified stratification by the buoyancy frequency averaged over the upper 300m water column (see **Methods**). Higher values of buoyancy frequency indicate a stratified water column where exchange from nutrient-rich water below the surface is reduced." (**Lines 139-142**)

We have also updated buoyancy frequency calculations to account for the top 300m. This is more accurate since all our occupations were in water with MLD less than 300m.

Reviewer #3 (Remarks to the Author):

Review of NCOMMS-20-46890

Seasonal mixed layer dynamics shape phytoplankton physiology, viral infection, and accumulation in the North Atlantic

Overview:

In this manuscript, Diaz and colleagues attempt to generalize how mixed layer dynamics in the Western North Atlantic impact phytoplankton physiology, using field data collected as part of the North Atlantic Aerosol and Marine Ecosystems Study (NAAMES). The authors attempt to link general markers of phytoplankton cell stress, including intracellular reactive oxygen species (ROS), cell membrane integrity, programmed cell death (PCD) protease activity, and lipid composition (oxidized phosphatidylcholine, triacylglycerols, and betaine-like lipids) to phases of the North Atlantic bloom cycle, thereby inferring physical mixing impacts upon cell physiology. In addition, the authors investigated changes in water column dissolved organic carbon (DOC), transparent exopolymer particles (TEP), and viruses. The authors investigate these parameters in the context of the disturbance-recovery hypothesis with the goal of using physiological markers and virus concentrations to better understand phytoplankton accumulation and loss. While I appreciate the authors' interest in linking large-scale physical mixing features with general markers of phytoplankton physiology, one major weakness of the study is the failure to acknowledge potential impacts of changing community composition upon the measurements and interpretations.

Bolanos et al. (2020) described significant community composition changes observed between different stations and seasons in the NAAMES study. Of particular concern is the shift from cyanobacterial dominated communities to those with >25% diatoms and prymnesiophytes. Using measured virus abundance as an example of how community composition could impact data interpretation, virus burst sizes depend upon the host. Viruses infecting larger eukaryotic phytoplankton tend to have burst sizes an order of magnitude larger than those infecting cyanobacteria. Could the changes in virus cell⁻¹ be explained by community composition changes?

We appreciate Reviewer#3 raising the importance of community composition in the interpretation of our findings. Also, as viral ecologists, we realize and appreciate the differences and impacts of host taxa on virus replication strategies, burst sizes, etc. We agree that this type of analysis can indeed provide important contextual insight into possible factors impacting our suite of intracellular and extracellular biomarkers, including the use and interpretation of virus concentrations/production. We have now included broad bloom-wide and phase-specific (climax and decline) PCA analyses of community composition using published data on 16S rRNA (Bolanos et al. 2021²) and HPLC pigments (Kramer et al. 2020³) as it relates to our markers (Fig. 7, Supplementary Fig. 10, 11). We refer Reviewer#3 to similar comments for Reviewers#1&4 regarding other general findings from those PCA analyses. With regard to viruses, only cyanobacteria (taken as the sum of Prochlorococcus, Synechococcus clades and other cyanobacteria) were most closely associated (co-varied) with higher virus concentrations and virus cell⁻¹ across all bloom-wide NAAMES stations (Fig. 7) and also within the Decline phase (Supplementary Fig. 11). Given virus production and associated lytic cell death would arguably reduce cell concentrations—this is the classic outcome of lytic infection— in combination with the fact that cyanobacteria have smaller burst sizes, we argue that the detected viruses do not derive from cyanobacteria (Lines 337-339).

We recognize that our use of SYBR staining to determine virus concentration is imperfect, given it does not detect the small (20-40nm capsid) ssRNA- or ssDNA-based viruses, known to infect diatoms. No flow cytometry-based technique to our knowledge has been developed for the detection of these viruses. Rather, molecular techniques targeting the RNA-dependent RNA polymerase (ssRNA) and replicate (ssDNA) genes in metatranscriptome/metagenome analyses have been used to detect these viruses. These techniques are difficult to use to accurately quantify virus particles, and they were not performed as part of this study. Consequently, our

flow-cytometry-based data paint a partial picture of how virus production changes with mixed-layer depths and across the seasonal North Atlantic bloom. We now provide discussion in the main text (**Lines 246-254**) providing context regarding the limitations of our virus cell⁻¹ measurements and diatoms, which were more abundant in the Climax phase (**Fig. 7**). Nonetheless, the fact that we have discernable patterns for the detectable viruses within biological systems across water masses and seasons is significant and revealing. Our observations of increased virus concentration and virus cell⁻¹ being positively correlated with stratification, subcellular death markers and accumulated DOC are novel and shed important insight into how microbial systems respond to MLD changes. We hope our work will inspire other scientists to apply omics techniques in similar systems to specifically identify which viruses are being produced throughout a bloom, along with the physiology of their respective hosts.

Similarly, how much would you expect community composition to impact TEP production?

Reviewer#3 raises an important point here. Our bloom-wide (**Fig. 7**) and intra-Climax phase (**Supplementary Fig 10**) PCA analyses mentioned above allowed us to address this comment regarding the impact of community composition on TEP cell⁻¹. We've added the following discussion to the main text: "Integrating community composition revealed some novel associations between biomarkers and community type. TEP cell⁻¹ notably clustered with communities in which diatoms were in higher relative abundance by either pigment or 16S community composition (**Fig. 7c, 7d, Supplementary Fig. 10c**). Some diatom species are known to have significantly higher TEP cell⁻¹ compared to other phytoplankton⁶⁶, so they may have been the source of the TEP. 16S-based community composition showed that *Micromonas* and *Cryptophyceae* also positively covaried with TEP cell⁻¹, implying that a community type, rather than one group of phytoplankton, are associated with high TEP cell⁻¹" (**Lines 320-328**).

With regards to physiological states, how universal are the markers used (oxidized phosphatidylcholine, triacylglycerols, betaine-like lipids, caspase and metacaspase activity) to the diverse assemblage of phytoplankton observed over the spatial and temporal scale of the study?

This point raised by Reviewer#3 questions the universality of the suite of lipid and protease activity markers. Reviewer#1 had a similar comment to which we provide a detailed response above, including a discussion on the broader interpretive context of our markers as they relate to the collective microbial system that is experiencing changes in mixed layer depth across the seasonal bloom. We argue that the fact that our measurements derive from a variety of microbial sources within the community strengthens and expands our central message regarding physiological state changes of a "biological system" within the mixed layer throughout a bloom. A main point from our study is that the system itself is characterized by discernable physiological states related to stratification throughout the bloom.

We've updated the main text with clearer descriptions of what types of organisms the biomarkers are informing us about and the size fractions we used to capture them (**Lines 160-180**). We also now include a table (**Supplementary Table 1**) as a guide for the specificity of these biomarkers and to clarify the organism types from which our various biomarker measurements (both cellular and environmental/ecosystem) likely derive and what they represent. It includes reference to the published literature on these biomarkers and aims to help the reader understand the likely processes that are connected to our measurements and provide context for our conclusions. For example, the presence (and activation) of caspase and metacaspase activities has been extensively documented in diverse eukaryotic phytoplankton taxa (diatoms, coccolithophores, chlorophytes, dinoflagellates, etc.) in response to cellular stress and in association with well-characterized cell pathways (Bidle 2015², 2016³ and references within). Oxidized PCs are less well documented in phytoplankton but, given the widespread occurrence

of PC as cellular lipids and their association with our ROS markers, we argue that they are robust lipid markers of oxidative stress in eukaryotic phytoplankton. TAGs are likely not produced by cyanobacteria so they most likely derive from eukaryotic cells, who use them as common energy storage lipids. It is important to note, however, that we do not know the exact taxa that are responsible for TAG ChIA⁻¹, OxPC total PC⁻¹, and caspase/metacaspase activity in our study. We also are unable to discern from whom viruses and DOC derive. Our data does show the type of community it may originate from, which can guide future efforts to understand bloom-driven physiology. For example, the covariance between high TAG ChIA⁻¹ and cyanobacteria in our PCA analysis (**Fig. 7c, 7d, Supplementary Fig. 10d, 11c, 11d**) does not mean these lipids are originating from cyanobacteria, given there have been no reports of marine cyanobacteria producing them. Rather, we think they are more likely from eukaryotic heterotrophs or eukaryotic algae that are experience cellular stress/death responses in Climax and Decline phases.

We've now added a paragraph discussing our interpretations of cyanobacteria communities associated with decline phase biomarkers (**Lines 327-343**) and diatoms and other members associated with high TEP cell⁻¹ (**Lines 320-327**). We've added an introductory paragraph before we go into our biomarker results clarifying the range of organisms that each biomarker likely covers, as well as how our biomarkers link the microbial community to stress and death (**Lines 160-180**). We've also added a note later in the main text that future research using more resolved separation and analytical techniques, including cell sorting and nucleic acid-based analysis, is needed to determine which phytoplankton are experiencing which stresses (**Lines 340-343**).

To address the extent to which these biomarkers vary over the spatial and temporal scale of our study, we've added a new analysis (**Fig. 6**), which plots the median values per station of samples within the mixed layer, underneath a map of the study area's sub-regions and station locations. We discuss in the main text that stations clustered together despite the geographic and macronutrient variability of each bloom phase, and that some biomarkers cluster together throughout the bloom. (**Lines 279-294**).

Specific comments:

Line 76-82 - All references cited are for eukaryotic phytoplankton, either include cyanobacteria, or be more specific about type of phytoplankton these processes apply to and relevance to ecosystem studied.

*Thank you for this comment. We have updated the references cited in the main text and include a guide for the specificity of biomarkers used in the study (**Supplementary Table 1**). Where appropriate, we have included discussion and/or reference to studies with cyanobacteria. As noted in the table, smaller cyanobacteria (<1.2 μm) were excluded based on our sampling setup for most of our biomarkers.*

Lines 259-261 - "The sudden transition to lower irradiance may have triggered some phytoplankton to undergo stress-induced viral infection and TEP production". The only citation is for a recent study on temperate viral infections in *Emiliana huxleyi*, but based on Bolanos et al. (2020), it is unlikely these were the dominant member of the phytoplankton community. Do you have other evidence that lysogenic viruses enter lytic phase upon light limitation?

*We appreciate Reviewer#3 for pointing this out. We did indeed reference a recent study on temperate infections in *E. huxleyi* for possible evidence of stress-induced production of viruses. It is unknown whether low light triggers the lytic induction of temperate infections in other eukaryotic algae. Given temperate infection has not yet been documented in other marine phytoplankton taxa and is more speculative, we have removed mention of low light induction.*

Also, Reviewer#3 is correct that comprehensive analysis of NAAMES community composition as performed Bolanos et al. 2021² and Bolanos et al. 2020⁶ showed that

chlorophytes were the dominant member in based on 16S rDNA. We've updated our reference from Bolanos 2020 to Bolanos 2021 since we use the same data from the Bolanos 2021.

Also, would you expect less TEP production with lessening of irradiance?

High light⁷ has been found to increase TEP production based on previous studies, as does general increases in phytoplankton concentration, which can increase in response to higher daily irradiance. We've updated the text to reflect these concepts. (Lines 369-376)

Lines 266-269 - TEP and virus concentrations in the surrounding waters increased consistent with nutrient stress and/ or viral infection. Wouldn't re-stratification also increase irradiance exposure, a potential cause of TEP production?

We appreciate Reviewer#3 for pointing this out. A confounding aspect of these data was that the TEP L⁻¹ increased, while the TEP cell⁻¹ decreased when going from deeply mixed to shallower mixed layers (Fig. 8, Supplementary Fig. 12). Several studies have shown that total TEP concentration scales with phytoplankton concentration/ChlA⁷⁻¹⁰, so we think that the increase in TEP L⁻¹ resulted from an increase in cell concentration (Fig. 8), in response to higher daily irradiance exposure in stratified water columns. We have reworded our discussion to indicate that the increase in TEP L⁻¹ was linked to an increase in overall phytoplankton or higher daily irradiance. We also note that Burns et al (2019)¹¹ found that TEP cell⁻¹ increased via turbulence-induced aggregation, so we suggest that TEP cell⁻¹ was higher in the deeply mixed condition because it was more turbulent and allowed TEP or TEP precursors to aggregate. We also suggest that the higher TEP cell⁻¹ may have been derived from bacterial colonization¹² of detrital cells found below the mixed layer, brought to the surface during the mixing event. (Lines 369-376)

Lines 271-274 - Seemed highly speculative.

We agree that this is speculative and have now removed this from the discussion. We attributed the combination of viruses increasing along with phytoplankton to "be indicative of a subset of phytoplankton or bacteria lysing/releasing viruses into the surrounding water in response to stratification." (Lines 358-361).

Lines 289 - 310 - "compared two water masses in the decline phase" - How did the communities compare between the subpolar (Decline Station 6) and subtropical (Decline Station 2)?

We've now added analyses which incorporates pigment-based and 16S rRNA-based community composition analysis with our biomarkers. It now reads: "These differences in nutrient concentration between these sub-regions in Decline phase may have shaped community composition and its impact on the aforementioned biomarkers (TAG, ROS, OxPC). HPLC-based pigment analysis showed that Decline Station 6 and Decline Station 2 were respectively haptophyte and cyanobacteria community types for both sampling dates. Cyanobacteria were still the most prominent group in Decline Station 6 based on 16S rRNA analysis but they were less dominant than at Decline Station 2 (30, 32% vs. 86% R.A, respectively). Decline Station 6 contained more Bacillariophyceae (diatoms) (13% vs. 0.1% R.A.) and Prymnesiophyceae (haptophyte group) (~13% vs. 1% R.A.)." (Lines 406-414).

Line 320 - "prolonged stratification" - what is the time scale for "prolonged"?

We were referring to the case study of Climax Station 4 (Figure 8), where we observed resident phytoplankton communities for two sampling days after being stratified, and then for an additional 5 days with our incubations. We updated the text to reflect this. (Lines 352-354)

Lines 325-326 - "These decline phase biomarkers have been extensively linked to cell death and late stages of viral infection in diverse model systems in culture". Metacaspase orthologs

have not been found in marine *Synechococcus* or *Prochlorococcus* (Bidle, 2015 and references within). The manuscript considered here should include discussion on how the relative abundance of these non-diazotrophic cyanobacteria might impact biomarker comparison among stations and the resulting interpretations of community physiology.

*We agree that this is an important point to note, that some of the stress/death biomarkers—e.g., caspase activity, metacaspase—have not been found in prominent marine unicellular cyanobacteria, such as *Synechococcus* or *Prochlorococcus*, so they would not contribute to these signals based on what's known about these groups. We have now included **Supplementary Table 1**, which gives examples describes our measured biomarkers, what they represent and which types of organisms they may derive from (with references). This general issue was also brought up by Reviewer#1 so please see corresponding response. A detailed discussion is now added to the main text (**Lines 160-180**).*

*To address how phytoplankton community composition may affect our biomarkers, we have incorporated data from other lab groups who used 16S rRNA² and pigments³ to construct a community profile throughout NAAMES. Our data showed that cyanobacteria positively covaried with TAG and stratification, and negatively covaried with nutrient concentration on a bloom wide and even intra-bloom scale (**Fig. 7**) and (**Lines 299-306**).*

*We highlight the fact that cyanobacteria also positively covaried with stratification, TAG ChlA⁻¹ and other biomarkers within the Decline phase (**Supplementary Fig. 11c, 11d**) with discussion in the main text. With this new community data incorporated, we discuss in further detail our reasoning as to why the caspase activity, TAGs, and other biomarkers, which positively covaried with cyanobacteria, did not likely derive from cyanobacteria, but rather from eukaryotic phytoplankton or grazers. We have updated the text to include an extension discussion of this line of reasoning (**Lines 327-343**).*

Methods:

Lines 379-397 - Viruses were quantified by flow cytometry, but it is often difficult to distinguish bacteriophage from instrument noise. I would like to see an illustrative flow cytometry plot in the supplementary information.

*Reviewer#3 is correct that virus concentrations were determined by flow cytometry using an BD Influx Mariner flow cytometer and that small viruses (~20-40 nm), which include bacteriophages, can sometimes be difficult to distinguish from the instrument noise. Our lab routinely performs these measurements on large dsDNA viruses (*Emiliana huxleyi* viruses/ EhV's) using SYBR-stained buffer blanks to determine the instrument noise and subtract it from samples with overlapping particle sizes. We now include **Supplementary Fig. 7** which shows our gating and blank correction strategy with a sample example from NAAMES. We note that more particles appear in the same size range in field samples compared to the background noise (blank samples) but that most of these particles do not contain green fluorescence and fall outside of the virus gate. Hence, they are excluded from the analysis. Consequently, our virus concentrations likely underestimate the true virus concentrations. We now provide discussion in the main text (**Lines 246-254**) regarding the limitations of our virus cell⁻¹ measurements (including how they relate to diatom viruses). Despite the technical limitations, the fact that we have discernable patterns for the detectable viruses within biological systems across water masses and seasons is still significant and revealing.*

I have concerns about viruses and TEP accumulation calculated from deck-board incubations while plankton accumulation rates were calculated from 5m depth in situ sampling. The authors note that the 5m depth phytoplankton accumulation rates were not statistically different from the in situ accumulation rates (Fig. 1c, 1d), but it seems simpler to use the same treatments for accumulation rates of phytoplankton, viruses, and TEP. Is there a compelling reason to use the 5m depth values?

We understand Reviewer#3's request/desire to measure virus and TEP accumulation using time-resolved samples from in situ water collected at different depths over the course of occupation. We agree that this would have been desirable but there were several logistical problems that made this unfeasible.

First, our lab group does not have the same time resolved data for in situ values of TEP and viruses/, as were collected for phytoplankton accumulation via flow cytometry. The latter measurements were only performed on water collected from 5 m, which limits coverage through the water column. They were taken whenever possible and were easily incorporated into the existing flow cytometry workflow. Since we wanted to understand how the cell physiology of populations throughout the mixed layer changed over bloom states, we included samples from a range of depths at each sampling time point (40%, 20% and 1% irradiance levels) for our analyses. This also provided much more repetition and statistical confidence.

Second, we were often only on site at stations for 1-2 days prior to moving along on our transect. For any station with only one day of occupation, we were not able to calculate any accumulation of any biomarker. Consequently, our strategy was to capture source water for 48-72 h, 8L on-deck incubations at in situ temperature and irradiance levels to capture and quantify accumulation of these parameters. The fact that we observed agreement in our phytoplankton accumulation rates between these incubations and in situ collected samples argue that this approach did not introduce significant artifacts or bias.

Lastly, our sampling strategy was critically informed by a previous study we conducted in the North Atlantic that examined TEP and virus infection¹³. That study was investigating the linkage between virus infection, TEP production, particles, and export fluxes. It also utilized similar on-deck incubations. One interesting observation was that there was a notable discrepancy between TEP accumulations made from in situ water collected via CTD/Niskin bottles and those made from incubation bottles. The former showed not accumulation while the latter showed notable accumulation. These disparate findings were interpreted as TEP being vertically exported over the course of daily in situ sampling via the CTD due to aggregation and sinking. Hence, TEP was actively removed, masking the accumulation. Incubation bottles didn't have this issue given the accumulating TEP was contained and could be measured.

Figures:

A map of transects and water masses would be an excellent Supp. Fig to orient the reader (e.g. Fox et al. 2020 Fig. 1)

We thank Reviewer#3 for suggesting this. We agree that this will be useful to orient the reader. We have updated **Fig. 1a** to show the locations of stations sampled during the NAAMES expeditions in relation to water types (Gulf Stream and Sargasso Sea, Subtropical, Temperate and Subpolar) as defined by Della Pena and Gaube 2019¹⁴. We have also added a new **Supplementary Fig. 1** that shows the tracks of profiling floats within our study site overlaid with water mass types. We do not show the actual cruise tracks from the four individual expeditions, given that our lab group did not sample the entire cruise track. We feel that this would have made the figure unnecessarily busy for the reader and that showing actual locations of where we sampled water masses through the four seasonal bloom phases was more effective. We have referenced the NAAMES overview article (Behrenfeld et al 2019¹⁵) which also presents the cruise transects.

References:

1. Sverdrup, H. U. On conditions for the vernal blooming of phytoplankton. *ICES J. Mar. Sci.* (1953) doi:10.1093/icesjms/18.3.287.
2. Bolaños, L. M. et al. Seasonality of the Microbial Community Composition in the North Atlantic. *Front. Mar. Sci.* **8**, 23 (2021).
3. Kramer, S. J., Siegel, D. A. & Graff, J. R. Phytoplankton Community Composition

- Determined From Co-variability Among Phytoplankton Pigments From the NAAMES Field Campaign. *Front. Mar. Sci.* **7**, (2020).
4. Bidle, K. D. The Molecular Ecophysiology of Programmed Cell Death in Marine Phytoplankton. *Ann. Rev. Mar. Sci.* **7**, 341–375 (2015).
 5. Bidle, K. D. Programmed Cell Death in Unicellular Phytoplankton. *Current Biology* vol. 26 R594–R607 (2016).
 6. Bolaños, L. M. *et al.* Small phytoplankton dominate western North Atlantic biomass. *ISME J.* **14**, 1663–1674 (2020).
 7. Claquin, P., Probert, I., Lefebvre, S. & Veron, B. Effects of temperature on photosynthetic parameters and TEP production in eight species of marine microalgae. *Aquat. Microb. Ecol.* **51**, 1–11 (2008).
 8. Passow, U. Production of transparent exopolymer particles (TEP) by phyto- and bacterioplankton. *Mar. Ecol. Prog. Ser.* **236**, 1–12 (2002).
 9. Corzo, A., Morillo, J. & Rodríguez, S. Production of transparent exopolymer particles (TEP) in cultures of *Chaetoceros calcitrans* under nitrogen limitation. *Aquat. Microb. Ecol.* **23**, 63–72 (2000).
 10. Lee, J. H. *et al.* Transparent exopolymer particle (TEPs) dynamics and contribution to particulate organic carbon (POC) in Jaran bay, Korea. *Water (Switzerland)* **12**, 1057 (2020).
 11. Burns, W. G., Marchetti, A. & Ziervogel, K. Enhanced formation of transparent exopolymer particles (TEP) under turbulence during phytoplankton growth. *J. Plankton Res.* **41**, 349–361 (2019).
 12. Busch, K. *et al.* Bacterial Colonization and Vertical Distribution of Marine Gel Particles (TEP and CSP) in the Arctic Fram Strait. *Front. Mar. Sci.* **4**, 166 (2017).
 13. Laber, C. P. *et al.* Coccolithovirus facilitation of carbon export in the North Atlantic. *Nat. Microbiol.* **3**, 537–547 (2018).
 14. Della Penna, A. & Gaube, P. Overview of (sub)mesoscale ocean dynamics for the NAAMES field program. *Front. Mar. Sci.* **6**, 384 (2019).
 15. Behrenfeld, M. J. *et al.* The North Atlantic Aerosol and Marine Ecosystem Study (NAAMES): Science motive and mission overview. *Front. Mar. Sci.* **6**, (2019).

Reviewer #4 (Remarks to the Author):

In the current manuscript by Diaz and colleagues, authors aim to link the effects of surface water stratification to phytoplankton dynamics and physiology within the same layer. By investigating water mixing by “season” along with phytoplankton abundance, oxidative stress levels (proxied by ROS concentration, membrane integrity, OxPC, and TEP), TAG and BLL levels, and caspase and metacaspase activity, authors were able to suggest an intimate connection between cell stress, death, and viral predation. Authors go further to suggest the current findings support the disturbance-recovery hypothesis.

As I was invited specifically for expertise in lipidomics, I'll focus the remaining comments on this portion of the text:

The Bligh and Dyer method of lipid extraction along with normal phase LCMS-MS are standard techniques for identifying lipids. While RT and fragmentation pattern matching are acceptable routes towards identification of a reasonable level, with so few lipids discussed in the manuscript I was surprised that no Supplementary figures or tables were included relating to

their identification. Authors report that they will include this raw data via GitHub after publication but it would be sufficient to say that these data could be reasonably displayed in a single figure (in the supplement) to convince the reader that the appropriate identifications were made.

*We appreciate Reviewer#4 pointing this out. We noticed that we made a mistake in the lipid methods reported. We now and have included both the correct Bligh and Dyer lipid extraction (Popendorf et al., 2013) and the correct high-performance liquid chromatography / electrospray-ionization high-resolution accurate-mass mass spectrometry analytical methods (Collins et al., 2015; Becker et al., 2018; 2021). We have now included a **Supplementary Fig. 4** which shows the MS2 spectra of representative TAG, PC, and oxidized PC. We also show the diagnostic retention times and m/z patterns of the representative PC and oxidized PC along with corresponding fragmentation patterns.*

Additionally, in figure 3a, there appear to be two replicates for the winter transition that could easily be statistical outliers. Was this addressed statistically and what would be the interpretation for why two replicates had such drastically high concentrations (~4x) OxPC relative to other samples of the same season and is there a possible explanation as to why there were no other replicates like this for the other seasons? Other related questions could be asked of the potential outliers in 3b, 3c, and 3d.

We do not know the reason for the high concentrations of OxPCs in some of our sample replicates; other biological replicate samples from the same water mass did not show the outlier high values. Given the replicate samples within water masses and within a particular cruise were all extracted and analyzed at the same time, in the same manner, and with the same instrument settings, it is unlikely due to differences in extraction/processing procedures or different instrument calibration. Based on this reasoning, we decided to include the outlier samples. Since we have a large number of sampling points per season, we don't think that these outliers are driving the statistical differences we found between cruises.

A final unrelated comment: there is very little discussion related to the breakdown of phytoplankton species/groups present during sampling (i.e. ratio of dinoflagellates to diatoms to bacteria, etc., does a single species dominate at a specific sample point or did a bloom occur during a specific sampling event). It could be interesting in terms of interpretation to discuss any of these events in relation to the trends observed in figure 5.

*We appreciate this suggestion. Reviewers #1 and #3 also commented on including some data and analyses on community composition (see description and responses above). We have now added extensive community composition analyses (**Fig. 7, Supplementary Fig. 10, 11**), using data from two independent methods (16S rRNA sequencing via Bolanos et al 2021 and HPLC analysis of pigments via Kramer et al 2020) to better characterize how community structure related to our biomarkers.*

*We analyzed if and to what degree our biomarkers covaried with community composition across all of the bloom phases (**Supplementary Figure 9**) and also across populations within either the Climax and Decline phases (**Supplementary Figures 10, 11**). Our bloom-wide, global analyses found that cyanobacteria taxa positively covaried with higher levels of stratification, virus cell⁻¹, TAG ChlA⁻¹, caspase activity, compromised membranes and DOC_{SA}; diatom taxa positively covaried with phosphate and nitrate/nitrite levels, TEP cell⁻¹, and oxidative stress markers, such as ROS and OxPC (**Supplementary Figure 8**). Given TAGs are not known to be a major component of marine cyanobacteria biomass and given enhanced caspase activity, virus production and cell death would tend to reduce cell concentrations, these findings support stratification-induced stress of eukaryotic taxa. Furthermore, the fact that oxidative stress and TEP cell⁻¹ markers clustered with communities in which diatoms were present (but not the only members) and were associated with relatively high nutrient concentrations argues against their induction via nutrient stress. We now discuss this in the main text. (**Lines 295-343**)*

We were unable to compare ratios of taxonomic groups (e.g., dinoflagellates and diatoms) to our other quantitative data such as bacteria concentrations for several reasons. First, the pigment-based community method used by Kramer et al 2020 provided a qualitative analysis of specific taxonomic groups/ “community type” per sample. It is not a quantitative technique whereby accurate concentrations of phytoplankton taxa are discernable. Second, the 16S rRNA-based community data is based on relative abundance of plastid V1V2 amplicons, without normalization for biomass or cell counts. Since eukaryotic phytoplankton contain a wide range of plastid copy numbers between and within taxonomic groups, it is also not possible to obtain accurate cell concentration numbers to normalize against bacteria or TEP. We now discuss how to improve future sampling and analysis efforts to better identify which taxa are experiencing stresses. It reads: “Additional size fractions, high speed cell-sorting, and/or single-cell analyses, combined with nucleic acid-based techniques would provide finer resolution on which taxa experienced specific physiological changes and paint a more comprehensive picture of virus infection dynamics throughout the North Atlantic bloom.” (**Lines 340-343**)

For the in-depth station comparisons in **Fig. 9**, we included data from each community composition method in the main text (**Lines 388-395, Lines 406-414**). **Supplementary Figs. 10 and 11** show how each station covaried with respect to taxonomic groups as well as our biomarkers, using the two separate community profiling methods mentioned above. We did not go into species-level detail, but rather we matched the same level of taxonomic resolution used by each community analysis as the original authors published (Kramer et al 2020, Bolaños et al 2021). Taxonomic groups with stronger associations at one station had a larger arrow pointing in that direction of the station.

We did not analyze if one species was blooming at particular sampling points, primarily given the community analyses used did not have the level of resolution. We also didn’t occupy stations long enough to show a temporal succession of species (we only had 1-2 days of in-situ occupation and sampling on site). We noted that a viral bloom/ lysis event may have occurred during occupation at Decline Station—due to the increase in compromised membranes, viruses per liter, and viruses per cell (**Supplemental Fig. 14**)—but we cannot discern which species were infected and removed. Such an event may be indicative of a subset of phytoplankton or bacteria lysing/releasing viruses into the surrounding water in response to stratification. We have included some discussion about (**Lines 340-343**) about additional sampling and analysis techniques that can be used in future studies to help disentangle these aspects.

References

- Popendorf, K. J., Fredricks, H. F. & Van Mooy, B. A. S. Molecular ion-independent quantification of polar glycerolipid classes in marine plankton using triple quadrupole MS. *Lipids* **48**, 185–195 (2013).
- Bolaños, L. M. et al. Seasonality of the Microbial Community Composition in the North Atlantic. *Front. Mar. Sci.* **8**, 23 (2021).
- Kramer, S. J., Siegel, D. A. & Graff, J. R. Phytoplankton Community Composition Determined From Co-variability Among Phytoplankton Pigments From the NAAMES Field Campaign. *Front. Mar. Sci.* **7**, (2020).

Reviewer comments, second round review:

Reviewer #2 (Remarks to the Author):

In response to my review the authors removed most of their attempts to link the biological markers with specific physical processes. They made the paper more readable by defining terms where appropriate. I have a few relatively minor comments as noted below.

In my original review, I noted that the term "water masses" was undefined in the text. In their response, the authors stated that they use the term "water mass" to refer to "water associated with each station sampled". Two points: First, this term is still undefined in the text. Second, this is a very unusual definition of the term "water mass" which usually refers to a volume of water with similar properties. I believe that the non-standard use of this term led to some confusion for the other reviewers too, and it would likely confuse readers. I would recommend that the authors replace this term with something more specific throughout the text, unless they are referring to a known water mass (e.g. eighteen degree water).

Throughout the paper the authors use "correlate" and "covary" interchangeably. While these terms can be synonymous, I would suggest using one consistently throughout the paper, unless the authors are implying a distinction between the words which should then be explained. Also, in some places "positive" is used in front of these terms, and in other places it is omitted (presumably with positively being implied). Again, either of these choices would be ok, but the usage should be consistent.

line 239: remove "higher". After all, if viruses and DOC are positively correlated with stratification, low virus and DOC should correspond to weak stratification (on average). If the authors are trying to say that viruses and DOC only correlate with stratification when the stratification is high in some sense, then this needs to be explained.

line 311: "associated" doesn't sound like the right word here. To me "associated" implies something qualitative, but I think the authors mean "correlated" or "covary" (see point above).

paragraph starting on line 344: Here the authors describe a "convective mixing event", but they don't show that convection is active. Given the depth of the mixed layer it seems likely that there was active convection (as opposed to wind-driven mixing or advection of mixed water), but this needs to be shown. Otherwise, simply refer to this as a deep mixed layer.

line 361: "in response to stratification". It sounds like the authors are implying a causal connection that they haven't shown. Something like "during periods of strong stratification" would be more appropriate.

paragraph starting on line 355: In this paragraph, the authors refer to a "transient" and "episodic" deep mixing event and a "rapid stratification event". These terms suggest events associated with physical processes. As noted in my original review, these events could equally be associated with lateral advection. I would suggest changing the wording from, for example, "rapid stratification event" to something like "the period with rapidly increasing stratification" which more closely follows what is actually being measured.

line 382: "simulated 5-day stratification of the mixed layer". What is being "simulated"? I can't find any explanation in the caption for Figure 8 which is referenced here.

line 482: In the first sentence of the conclusion section, the authors claim that "seasonal water column stratification regulates intracellular stress, programmed cell death...". The authors have shown that these processes are *correlated* with stratification, but in my view they haven't shown that they are *regulated* by stratification and this word should be changed.

Reviewer #3 (Remarks to the Author):

In "Seasonal mixed layer depth shapes phytoplankton physiology, viral production, and accumulation in the North Atlantic" by Diaz et al., the authors aim to characterize the in situ physiological state of North Atlantic Ocean plankton communities in various bloom stages using a suite of biomarkers to better understand cellular losses associated with water column stratification. This work is both relevant and timely as increased surface ocean stratification resulting from anthropogenic carbon dioxide emissions has both been documented and is predicted to increase.

In the revised manuscript, the authors present additional analyses, supplementary materials and clarifying text that significantly improve upon the original submission. The authors have adequately addressed my concerns with the initial version.

Reviewer #4 (Remarks to the Author):

As I was invited specifically for expertise in lipidomics, I've only commented on this portion:

The correction to the lipid extraction and analytical profiling techniques used, as well as the additional representative MS2 spectra (Supplementary figure 4) are well appreciated.

The response related to winter outliers was also acceptable. I do agree with the authors that due to high replication these questionable data points likely do not drive statistical analyses. Due to the already expanded text for a communication, I do not think it is necessary to expand upon these points further in the main text.

Lastly, reviewers 1 and 3 also touched upon the need for an expansion of the community composition. Their questions were more exhaustive and as such I'll leave commentary to them. However, I do appreciate the lack of resolution of much of this data and the constraints that this has put on further dissecting cause and effect related to individual taxonomic groups.

Final revisions for *Nature Communications* manuscript NCOMMS-20-46890A

Seasonal mixed layer depth shapes phytoplankton physiology, viral production, and accumulation in the North Atlantic" by Diaz et al.

Response to Reviewers (*in blue italics*)

Reviewer #2 (Remarks to the Author):

In response to my review the authors removed most of their attempts to link the biological markers with specific physical processes. They made the paper more readable by defining terms where appropriate. I have a few relatively minor comments as noted below.

We greatly appreciate Reviewer#2's comments and recognition of our efforts to remove specific links to dynamic physical processes and to make the paper more readable by defining terms for the reader. We address their specific concerns below.

In my original review, I noted that the term "water masses" was undefined in the text. In their response, the authors stated that they use the term "water mass" to refer to "water associated with each station sampled". Two points: First, this term is still undefined in the text. Second, this is a very unusual definition of the term "water mass" which usually refers to a volume of water with similar properties. I believe that the non-standard use of this term led to some confusion for the other reviewers too, and it would likely confuse readers. I would recommend that the authors replace this term with something more specific throughout the text, unless they are referring to a known water mass (e.g. eighteen degree water).

We understand Reviewer#2's concern about using the term 'water mass'. We have removed usage of the term "water masses" and replaced it with either "stations", "sub-region" (or something appropriate to that effect depending on the context of the sentence). This usage more accurately reflects our samples/analyses and avoids any confusion regarding non-standard definitions of 'water mass'.

Throughout the paper the authors use "correlate" and "covary" interchangeably. While these terms can be synonymous, I would suggest using one consistently throughout the paper, unless the authors are implying a distinction between the words which should then be explained. Also, in some places "positive" is used in front of these terms, and in other places it is omitted (presumably with positively being implied). Again, either of these choices would be ok, but the usage should be consistent.

We have now better clarified our use of these words. We replaced the word "correlate" with "covary" in instances where we were actually discussing PCA results, which should be referred to as "covarying". In the one case where we still use "correlate", it is in reference to a LOESS line, so we think this is still appropriate. We also put "positive" in front of the two instances where it was not used (Line 326-327), but did not put positive in front of another instance since the variable in question was neither positive nor negatively covarying (Line 312).

line 239: remove "higher". After all, if viruses and DOC are positively correlated with stratification, low virus and DOC should correspond to weak stratification (on average). If the authors are trying to say that viruses and DOC only correlate with stratification when the stratification is high in some sense, then this needs to be explained.

We agree and removed 'higher'.

line 311: "associated" doesn't sound like the right word here. To me "associated" implies something qualitative, but I think the authors mean "correlated" or "covary" (see point above).

We agree and replaced 'associated' with covary, since its referring to PCA.

paragraph starting on line 344: Here the authors describe a "convective mixing event", but they don't show that convection is active. Given the depth of the mixed layer it seems likely that there was active convection (as opposed to wind-driven mixing or advection of mixed water), but this needs to be shown. Otherwise, simply refer to this as a deep mixed layer.

We changed the verbiage here to just say deep mixed layer, as suggested.

line 361: "in response to stratification". It sounds like the authors are implying a causal connection that they haven't shown. Something like "during periods of strong stratification" would be more appropriate.

We edited the text as suggested. It now reads: *"Increased phytoplankton accumulation is likely due to the increased daily availability of light and reduced grazer presence⁶⁹, while increased viruses L^{-1} may be indicative of a subset of phytoplankton or bacteria lysing/releasing viruses into the surrounding water during periods of strong stratification". (Lines 416-419)*

paragraph starting on line 355: In this paragraph, the authors refer to a "transient" and "episodic" deep mixing event and a "rapid stratification event". These terms suggest events associated with physical processes. As noted in my original review, these events could equally be associated with lateral advection. I would suggest changing the wording from, for example, "rapid stratification event" to something like "the period with rapidly increasing stratification" which more closely follows what is actually being measured.

We removed verbiage describing a deep mixing event, and mainly replaced it with references to the mixed layer depth shallowing.

line 382: "simulated 5-day stratification of the mixed layer". What is being "simulated"? I can't find any explanation in the caption for Figure 8 which is referenced here.

We revised this sentence to specifically reference the incubations that were used. It now reads: *Given the general increase in stress and death markers for these populations after a five day incubation (Fig. 8), we expected cell populations at Climax Station 1 to resemble the communities at Climax Station 4 after stratification/incubation.* (lines 477-479).

line 482: In the first sentence of the conclusion section, the authors claim that "seasonal water column stratification regulates intracellular stress, programmed cell death...". The authors have shown that these processes are *correlated* with stratification, but in my view they haven't shown that they are *regulated* by stratification and this word should be changed.

We've changed this wording to "is correlated with."

Reviewer #3 (Remarks to the Author):

In "Seasonal mixed layer depth shapes phytoplankton physiology, viral production, and accumulation in the North Atlantic" by Diaz et al., the authors aim to characterize the in situ physiological state of North Atlantic Ocean plankton communities in various bloom stages using a suite of biomarkers to better understand cellular losses associated with water column stratification. This work is both relevant and timely as increased surface ocean stratification resulting from anthropogenic carbon dioxide emissions has both been documented and is predicted to increase.

In the revised manuscript, the authors present additional analyses, supplementary materials and clarifying text that significantly improve upon the original submission. The authors have adequately addressed my concerns with the initial version.

We appreciate Reviewer#3's recognition of the relevance and timeliness of our work, as it relates to predicted changes of increased ocean stratification in response to climate change. We are pleased that our inclusion of additional analyses, supplementary materials and text edits served to clarify and improve the original submission and adequately address their concerns.

Reviewer #4 (Remarks to the Author):

As I was invited specifically for expertise in lipidomics, I've only commented on this portion:

The correction to the lipid extraction and analytical profiling techniques used, as well as the additional representative MS2 spectra (Supplementary figure 4) are well appreciated.

We are pleased to hear that our corrections and additional Supplementary Figures were appreciated and helpful to clarify.

The response related to winter outliers was also acceptable. I do agree with the authors that due to high replication these questionable data points likely do not drive statistical analyses. Due to the already expanded text for a communication, I do not think it is necessary to expand upon these points further in the main text.

We are pleased that our responses adequately answered Reviewer#4's questions about outliers.

Lastly, reviewers 1 and 3 also touched upon the need for an expansion of the community composition. Their questions were more exhaustive and as such I'll leave commentary to them. However, I do appreciate the lack of resolution of much of this data and the constraints that this has put on further dissecting cause and effect related to individual taxonomic groups.

We appreciate Reviewer#4's comments regarding community composition and deferral to Reviewers#1 and #3 for specifics. Given neither reviewer had follow up questions or concerns about our inclusion of extensive analyses of phytoplankton community composition as it related to season mixed layer depth, we assume our edits sufficiently resolved their original questions/concerns.